# Recreation of an antigen-driven germinal center in vitro by providing B cells with phagocytic antigen

Ana Martínez-Riaño[1,3], Pilar Delgado [1,3], Rut Tercero[1,3], Sara Barrero[1], Pilar Mendoza[1], Clara L. Oeste [1], David Abia [1], Elena Rodríguez-Bovolenta[1], Martin Turner [2] & Balbino Alarcón [1✉]

Successful vaccines rely on activating a functional humoral immune response through the generation of class-switched high affinity immunoglobulins (Igs). The germinal center (GC) reaction is crucial for this process, in which B cells are selected in their search for antigen and T cell help. A major hurdle to understand the mechanisms of B cell:T cell cooperation has been the lack of an antigen-specific in vitro GC system. Here we report the generation of antigen-specific, high-affinity, class-switched Igs in simple 2-cell type cultures of naive B and T cells. B cell antigen uptake by phagocytosis is key to generate these Igs. We have used the method to interrogate if T cells confer directional help to cognate B cells that present antigen and to bystander B cells. We find that bystander B cells do not generate class-switched antibodies due to a defective formation of T-B conjugates and an early conversion into memory B cells.

---

[1] Centro de Biologia Molecular Severo Ochoa, CSIC-UAM, 28049 Madrid, Spain. [2] The Brabaham Institute, Babraham Hall House, Babraham, Cambridge CB22 3AT, UK. [3] These authors contributed equally: Ana Martínez-Riaño, Pilar Delgado, Rut Tercero. ✉email: balarcon@cbm.csic.es

For an efficient protective humoral response against pathogen-derived protein antigens, B cells establish an intimate collaboration with antigen-specific helper T cells. To obtain T-cell help, B cells have to recognize cognate antigen via their B cell antigen receptor (BCR), internalize the antigen, and present its processed form as MHC class II-associated peptides. CD4 T cells that are able to recognize the processed antigen will become activated and express ligands of costimulatory receptors for B cells that, in turn, will promote proliferation and somatic hypermutation[1]. These processes result in the selection of B cells bearing class-switched immunoglobulins of high affinity for the antigen. This B-T cell cooperation takes place in germinal centers (GC), where B cells undergo iterative cycles of antigen recognition and presentation to T cells, followed by very rapid cell proliferation and expansion. It is generally accepted that in GCs, B cells establish a fierce competition for the antigen to gain T cell help, resulting in the selection of B cells bearing BCRs with the highest affinity[2]. The BCR can interact and be activated by soluble proteins. Nevertheless, reports show that B cells more often recognize and take up antigens deposited on the surface of antigen-presenting follicular dendritic cells[3–5]. There is a long-standing belief that only antigen-presenting cells of myeloid origin are able to phagocytose antigens and that B cells are not competent to phagocytose particulate antigens[6,7]. However, we showed that follicular B cells can also acquire cognate antigens through phagocytosis[8].

B cells receive help from a type of activated helper CD4 T cell known as T follicular helper cells (Tfh). These cells release important cytokines that stimulate B cell proliferation and modulate Ig class switching, including IL4 and IL21. Furthermore, Tfh expresses ligands (CD40L, ICOS) for costimulatory receptors in B cells (CD40 and ICOSL)[9]. B cells integrate signals emanating from their antigen-engaged BCR, from ligated CD40 and ICOSL, as well as from cytokine receptors to promote their program of affinity maturation and Ig class switching. In this context, the BCR has a dual function: as a provider of activation signals to the B cell and as a mediator of antigen internalization, processing, and presentation to T cells. It is a challenge to distinguish the extent to which BCR function is mediated indirectly by its role in antigen presentation and signals transmitted by CD40 and other T cell-engaged costimulatory receptors, or mediated directly through BCR signaling[10]. One of the caveats for detailed studies on the molecular processes of B-T cell interaction is the inability to recreate GCs in vitro. Different protocols consisting of mixtures of cytokines and the expression of CD40L in non-T cells have been used[11]. However, these procedures are not antigen-specific and have not been successful in selecting Ig class-switched B cells with an increasing affinity for antigen.

Receptor-mediated phagocytosis of particulate material requires an actin-dependent zippering of the membrane around the particle, forming a cup that leads to progressive engulfment[12]. Phagocytosis is regulated by GTPases of the Rho family. According to the involvement of different Rho family members, phagocytosis is classified into two main groups. Type I phagocytosis involves Rac1 and Cdc42, such as that observed for the Fc Receptor (FcR). In turn, type II phagocytosis involves RhoA, as described for the Complement Receptor 3 (CR3; reviewed in ref. [13]). Another important GTPase is RhoG, which is an evolutionarily conserved, intracellular mediator of apoptotic cell phagocytosis[14,15]. Interestingly, in an RNA interference screen of 20 Rho GTPases in macrophages, RhoG was found to be required for particle uptake mediated by both FcR and C3R[16]. At odds with the idea that lymphocytes are not professional phagocytes, we previously found that RhoG was involved in the nibbling of MHC-associated portions of the membrane of antigen-presenting cells by T cells[17]. This process, known as trogocytosis, also requires the activation of another small GTPase, in this case of the R-Ras subfamily, known as R-Ras2 or TC21, which is a direct interactor of the T cell receptor (TCR). In addition, we found that T cells can phagocytose 1–6 μm diameter latex beads coated with anti-CD3 antibodies by an R-Ras2- and RhoG-mediated process[17]. The capacity of T cells to phagocytose particles by a TCR-driven mechanism is paralleled by a similar behavior of B cells. In fact, we recently showed that naive follicular B cells can phagocytose 1–3 μm beads coated with their cognate antigen[8]. This process is BCR-driven and leads to antigen presentation to T cells, which subsequently engage in several rounds of cell divisions[8]. Furthermore, this BCR-driven antigen phagocytosis efficiently generates germinal centers (GC) in vivo, producing antigen-specific class-switched Igs of high affinity. Having demonstrated the stimulatory effect of antigen phagocytosis by B cells on GC formation in vivo, we have now investigated its potential use for the recreation of antigen-driven GCs in vitro.

## Results

**In vitro system for generating antigen-specific class-switched antibodies of high affinity.** B cells require two signals to differentiate into GC B cells: a cognate BCR stimulation and the survival and mitotic signals obtained from Tfh cells. Here, we investigated the possibility of using antigen phagocytosis by B cells to recreate a GC in vitro. The general set-up of the system consisted of a co-culture of T cells from OT-2 TCR transgenic mice specific for peptide 323-339 of chicken ovalbumin presented by I-A^b [18] and nitrophenol (NP) hapten-specific follicular B cells from B1-8^hi knock-in mice[19]. B1-8^hi knock-in mice bear a rearranged VDJ region in the IgH locus that, in combination with a rearranging lambda light chain, confers specificity for the 4-hydroxy-3-nitrophenylacetyl (NP) hapten and its iodinated derivative 4-hydroxy-3-iodo-5-nitrophenylacetic acid (NIP). Both B and T-cell types were purified either by negative selection (Fig. 1) or by cell sorting to exclude the participation of a third cell type that could be contaminating the B and T-cell populations (Fig. S1). Before mixing with T cells, B1-8^hi follicular B cells were pre-incubated with 1 μm latex beads coated with NP-ovalbumin (NIP-OVA). The mixture of T, B cells, and NIP-OVA beads was incubated for 7 days and the culture supernatants were tested for the presence of anti-NP antibodies of different isotypes by ELISA (Fig. 1a). Control experiments were carried out in the same conditions but providing the NIP-OVA antigen in soluble form instead of bead-bound. When stimulated with bead-bound antigens, cognate B cells completely phagocytose them[8].

In order to set up the in vitro system and determine the effect of either soluble or bead-bound antigen, we first carried out a titration experiment in which the capacity of B cells that have taken-up NIP-OVA to present antigen to OT-2 T cells was tested. OT2 T cells proliferated in a dose-response fashion to a fixed number of B cells previously incubated with soluble or bead-bound NIP-OVA (Fig. 1b, plotted data for this and all Figures in Supplementary Data 2). From this experiment, we chose the soluble and bead-bound NIP-OVA amounts that elicited similar OT2 T-cell proliferation: three coated beads per B cell and 100 ng/ml of soluble NIP-OVA. The experiment repeated at these antigen doses showed that OT2 cells proliferated equally and also expressed the markers of follicular helper T cells (Tfh) CXCR5 and PD1 to the same extent (Fig. 1c). However, soluble antigen was significantly less mitogenic than bead-bound antigen for B1-8^hi B cells mixed with OT-2 T cells (Fig. 1d).

The production of antibodies of high and low-affinity towards the NP antigen after co-incubation of the B1-8^hi B cell/OT-2 T cell mix with either soluble or bead-bound NIP-OVA antigen was tested in the supernatant of 7-day cultures.

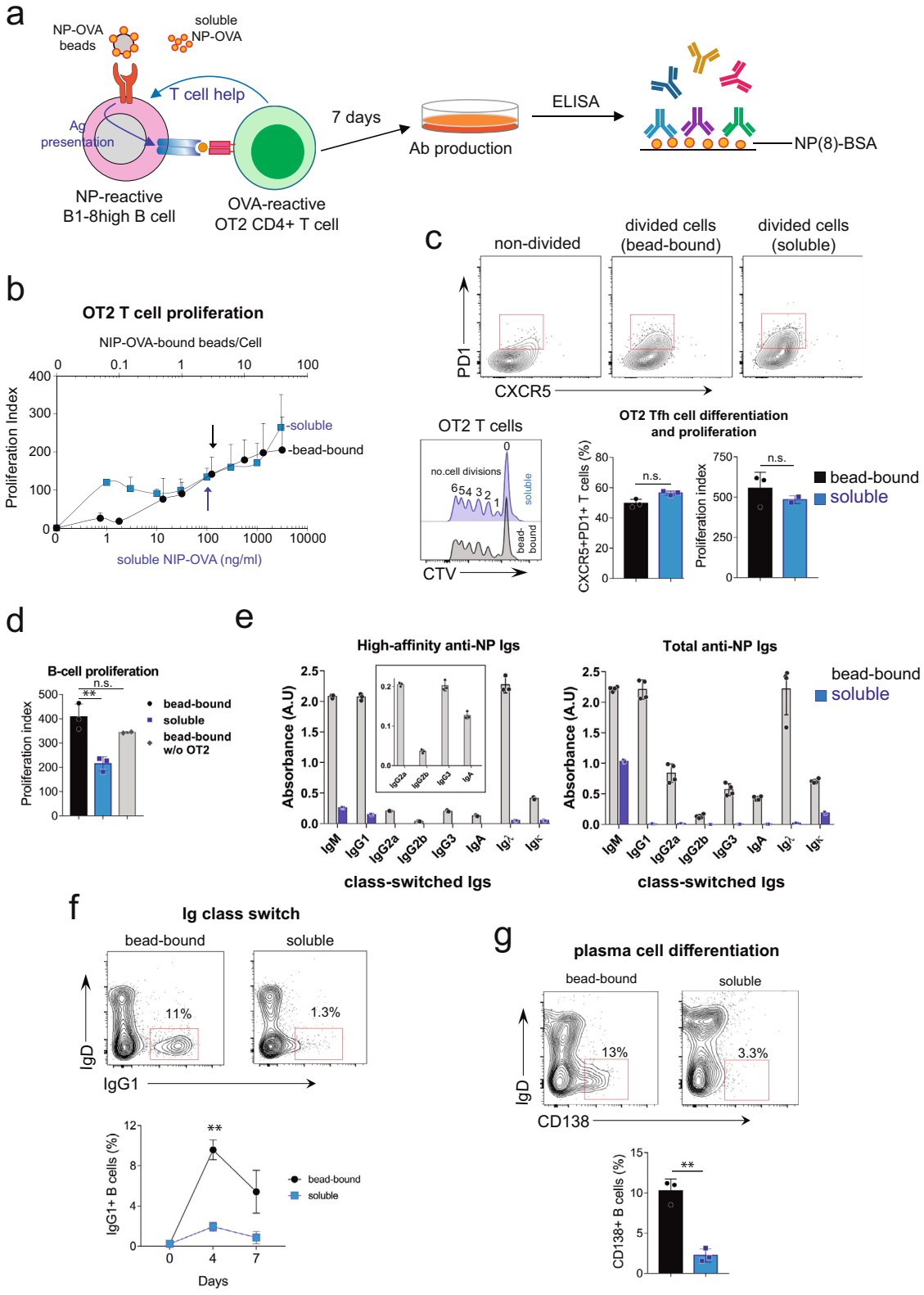

The supernatant of B1-8<sup>hi</sup> B cell cultures stimulated with bead-bound antigen, but not with soluble antigen, contained high-affinity class-switched Igs, including IgG1, IgG2a, IgG2b, IgG3, and IgA (Fig. 1e). By contrast, total and high-affinity IgM anti-NP was found in both types of culture, although more in the bead-bound one. In regard to the light chain, it appears that all heavy chains were preferably associated with Igλ and not Igκ, as previously described for NP-reacting antibodies[20]. The greater ability of bead-bound antigen to induce Ig class switch over that of soluble antigen was also evidenced by the expression of IgG1 at the B cell membrane (Fig. 1f). The higher production of antibodies by bead-bound antigen was in turn reflected by more cells bearing the plasma cell differentiation marker CD138 (Fig. 1g).

**Fig. 1 B cells make high-affinity antigen-specific class-switched immunoglobulins when cultured in vitro with antigen-coated beads and cognate T cells. a** Cartoon summarizing the general setting. NP-specific B1-8$^{hi}$ follicular B cells are incubated with 1 μm beads coated with NP-OVA, or with soluble NP-OVA, in the presence of OVA-specific OT-2 CD4 T cells for 7 days. The culture supernatant is then analyzed by ELISA for the presence of NP-specific antibodies. **b** Proliferation of OT-2 T cells after 4 days of culture with WT B1-8$^{hi}$ B cells stimulated at different doses of bead-bound (black circles) or soluble (blue squares) NIP-OVA antigen. Arrows indicate the bead/B cell ratio (3: 1) and soluble antigen concentration (100 ng/ml) that induce comparable OT-2 T cell proliferation. **c** Graph plots of Tfh marker (CXCR5 and PD1) expression and proliferation of OT-2 T cells after 4 days of culture with the antigen conditions selected in **b**. The inset illustrates how OT-2 T cell division is counted according to the peaks of CTV dilution. **d** Proliferation of WT B1-8$^{hi}$ B cells after 4 days of culture was calculated by CTV dilution after stimulation as in **b**. A control of B cells cultured with bead-bound NIP-OVA in the absence of OT-2 cells was carried out in parallel. Data represent the mean ± S.D. ($n = 3$). **$p < 0.01$ (unpaired Student's $t$ test). **e** Detection of high-affinity and total anti-NP Igs in supernatants of B1-8$^{hi}$ B cells stimulated with soluble (blue) or bead-bound (black) NIP-OVA together with OT-2 T cells for 7 days. High-affinity antibodies were measured by ELISA on plates coated with NP(7)-BSA and total antibodies on plates coated with NP(41)-BSA. Data represent the mean ± S.D. ($n = 3$). **f** Contour plots showing the appearance of Ig class-switched IgG1$^+$ IgD$^-$ B1-8$^{hi}$ B cells after 4 days in culture as in **b**. The line plot below shows the appearance of IgG1$^+$ cells on gated B220 + B cells after 4 and 7 days in culture. Data represent the mean ± S.D. ($n = 3$). **$p < 0.01$ (unpaired Student's $t$ test). **g** Contour plots showing the appearance of plasma cells (B220$^+$CD138$^+$ IgD$^-$) after 4 days in culture as in **b**. A quantification is shown in the bar plot below. Data represent the mean ± S.D. ($n = 3$). **$p < 0.01$ (unpaired Student's $t$ test).

**A bead-bound phagocytic stimulus provides a strong and sustained BCR signal.** To provide a mechanistic explanation for the findings above, we considered whether the bead-bound stimulus could result in a more intense or more sustained BCR signal than soluble antigen. We first determined the degree of occupancy of the BCR in both conditions using a fluorescent NP derivative. At the conditions used (3 coated beads vs. 100 ng/ml of soluble protein), 35% of the B1-8$^{hi}$ BCR was free to bind NP hapten in cells incubated with beads, whereas only 1% was free if cells had been incubated with soluble protein (Fig. 2a). We also measured surface BCR downregulation as a function of time and found that the soluble stimulus was at least as effective as the bead-bound stimulus at promoting BCR downregulation (Fig. 2b). Therefore, the reduced ability of soluble antigen to produce class-switched high-affinity antigen-specific Igs is not explained simply by lower BCR occupation.

Other signaling events downstream of the BCR were assayed. Phagocytosis requires the rearrangement of the actin cytoskeleton around the particle in the phagocytic cup[21]. We, therefore, tested the intensity and duration of actin polymerization in B cells incubated with bead-bound vs. soluble antigen. Both stimuli equally increased polymerized F-actin levels in B cells after 1 min of incubation (Fig. 2c). However, whereas the polymerization phase was rapidly followed by intense depolymerization in B cells stimulated with soluble antigen, the high F-actin content was sustained in B cells stimulated with bead-bound antigen. Phosphorylation of Akt and ERK, which are two events linked to activation of the PI3K and Ras pathways, were more intense and sustained with the bead-bound than with the soluble stimulus (Fig. 2d). More importantly, phosphorylation of Syk, a direct BCR effector previously shown to mediate FcγR- and CR-dependent phagocytosis[22,23], was also more intense and sustained upon stimulation with bead-bound antigen (Fig. 2d). These data suggest that the bead-bound stimulus induces a stronger and more persistent BCR signal than the soluble stimulus.

To determine if the stronger signal promoted by the bead-bound stimulus was related to the phagocytic process, we compared the phosphorylation of Akt and S6 (in the PI3K pathway) and of ERK in WT vs RhoG-deficient B cells in response to bead-bound antigen. RhoG is a GTPase that has been shown to be required for BCR-induced phagocytosis[8]. We found that RhoG is required to induce and sustain those signals, as well as the phosphorylation of Syk and of the Igα subunit of the BCR, strongly suggesting that antigen phagocytosis elicits a longer and more intense BCR signal (Fig. 2e). We next assessed the cellular location of phosphorylated BCRs during antigen phagocytosis. Using 1 μm fluorescent beads and confocal microscopy, we found that in B cells stimulated with bead-bound antigen for a short

time (5 min), both phospho-Igα and phospho-Syk were only detected in the phagocytic cups (Fig. 2f). Interestingly, both proteins remained phosphorylated all around the phagocytosed beads at a late (30 min) time point. These results show that BCR phosphorylation persists in the intracellular phagosome and suggest that this might be responsible for sustained BCR signaling when antigen is phagocytosed.

**Phagocytic B cells and T cells form organized structures in vitro.** In germinal centers, antigen-specific B cells form clusters of highly proliferating cells that segregate from non-responding B cells in follicles. In the culture plates in vitro, we found the formation of large clusters containing as many as 1700 cells when mixtures of NP-specific B1-8$^{hi}$ B cells and OT-2 T cells were incubated with 1-μm beads coated with NIP-OVA (Fig. 3a). Interestingly, stimulation of B cells with a similar dose of soluble NIP-OVA, resulted in the formation of much smaller clusters, suggesting that the large B cell and T-cell aggregates were related to the phagocytic stimulus. Using mixtures of CTV-labeled B1-8$^{hi}$ B cells and CFSE-labeled OT-2 T cells incubated with NIP-OVA antigen-coated beads for 7 days, we found that the periphery of the cluster contained the cells with most diluted CTV and CFSE, suggesting that B and T cells proliferate and expand towards the edges of the clusters (Fig. 3b). The periphery of the clusters also contained the highest percentage of B cells positive for the GC marker, GL7 (Fig. 3c), suggesting that B cells proliferate and express GC markers towards the periphery. These data show that follicular B cells stimulated with antigen-coated beads form large clusters, together with T cells, that are reminiscent of germinal centers.

**Phagocytic B cells from T-B co-cultures differentiate to GC B cells in vitro.** To determine if GC B-cell differentiation was induced upon phagocytosis in an antigen-specific manner, we incubated purified B1-8$^{hi}$ B cells with 1 μm beads coated with NIP-OVA in the presence of OT-2 T cells for 4 days. This led to the emergence of GC B cells characterized by the expression of the GL7 and CD95 markers and to B-cell proliferation (Fig. 4a, b). Beads coated with NP linked to a different carrier protein (chicken gammaglobulin, CGG) did not elicit GC B cell differentiation or proliferation, indicating that T-cell help is required. Likewise, the acquisition of a GC B-cell phenotype was inhibited if B cells lacked RhoG, suggesting that antigen phagocytosis was required for GC differentiation (Fig. 4a, b).

A key feature of GC B cell differentiation is the expression of the transcription factor Bcl-6 (Basso et al., 2012). We followed the emergence of a Bcl-6$^+$ B cell population that also downregulates

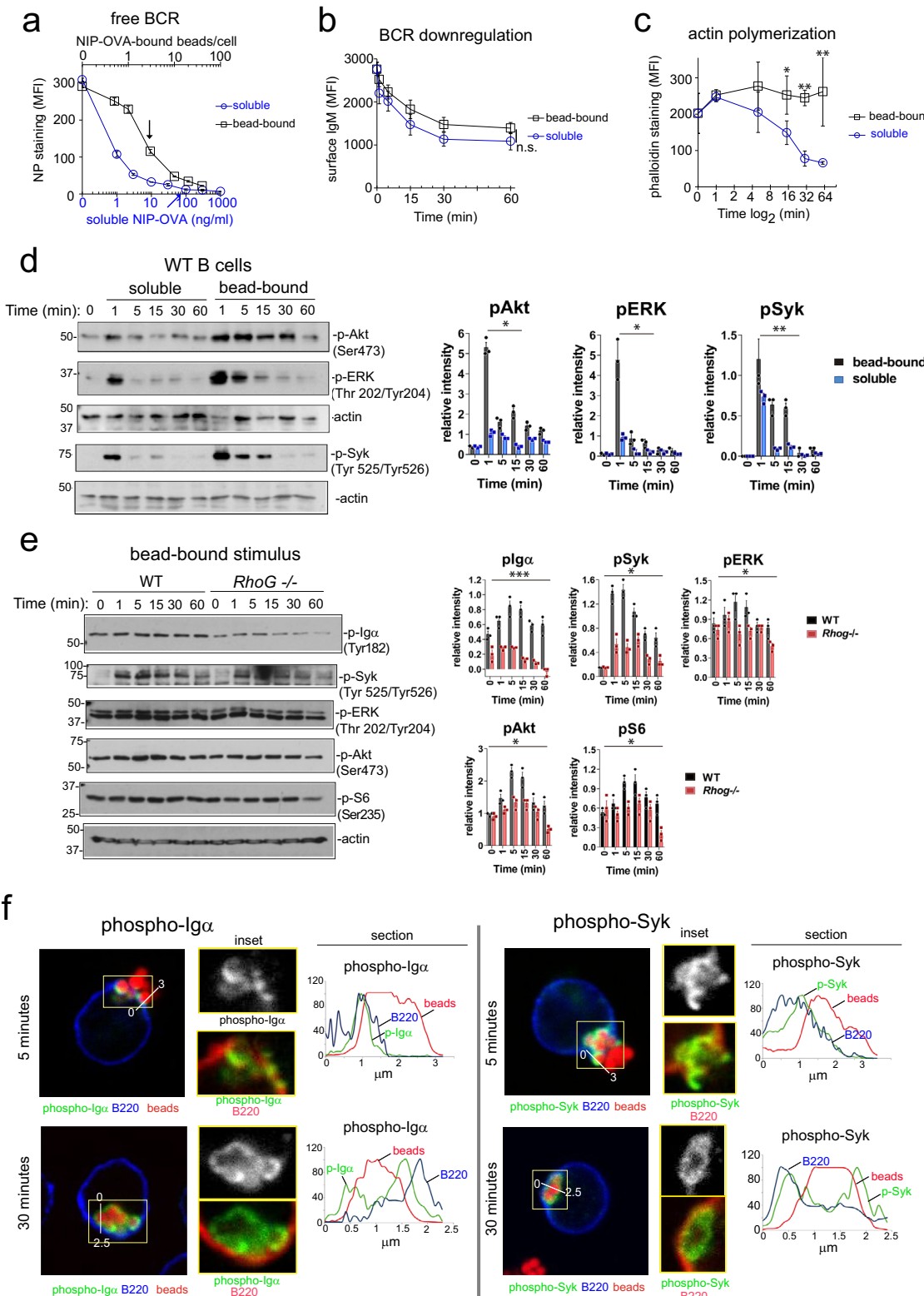

CD38 (both indicative of GC cells) during 7 days in a co-culture of WT B1-8[hi] B cells, NIP-OVA-coated 1 μm beads, and OT-2 T cells. We found a distinct Bcl-6[+]CD38[−] population indicating the differentiation of phagocytic B cells towards GC B cells (Fig. 4c). The maximum percentage of GC B cells was reached at day 5 and decreased thereafter. In parallel to GC markers, we followed the expression of the plasma cell transcription factor Blimp-1 and the surface marker CD138. Increased Blimp-1 was

detected at day 3, while the emergence of a distinct Blimp-1[+]CD138[+] plasma cell population was clearly evident at day 4, reaching a plateau at day 5 (Fig. 4c). Bcl-6 and Blimp-1 are connected in a regulatory feedback loop in which Bcl-6 represses Blimp-1 and vice versa, such that Blimp-1 expression favors exit from the GC reaction and differentiation to plasma cells (Basso and Dalla-Favera, 2010; Rui et al., 2011). We found that Bcl-6 and Blimp-1 are expressed reciprocally amongst gated B220[+]GL7[+]

**Fig. 2 A phagocytic antigen induces a stronger and more sustained BCR signal than a soluble one. a** Surface BCR saturation plot of purified B1-8[hi] B cells incubated with bead-bound (black) or soluble NIP-OVA (blue) antigen at different doses for 1 hour at 0 °C. Free unbound BCR was estimated by staining with NP-PE. Data represent the mean ± S.D. ($n = 3$). Arrows indicate the bead-dose and soluble concentration determined previously with comparable effects on OT-2 T cell proliferation (Fig. 1b). **b** BCR downmodulation was estimated according to anti-IgM staining of B1-8[hi] B cells after stimulation with bead-bound (black, 3:1 ratio) or soluble (blue, 100 ng/ml) NIP-OVA antigen for different time points at 37 °C. Data represent the mean ± S.D. ($n = 3$). **c** F-actin content was measured by phalloidin staining of B1-8[hi] B cells after stimulation with bead-bound (black, 3:1 ratio) or soluble (blue, 100 ng/ml) NIP-OVA antigen for different time points at 37 °C. Data represent the mean ± S.D. ($n = 3$). *$p < 0.05$; **$p < 0.01$ (unpaired Student's $t$ test). **d** Immunoblot analysis of phosphorylation events downstream of the BCR after stimulation of WT B1-8[hi] B cells with either a 3:1 ratio of bead-bound NIP-OVA or with 100 ng/ml of soluble NIP-OVA for different time points. Plots to the right show protein phosphorylation levels relative to the amount of actin quantified by densitometry. Relative density is shown as the mean ± SEM of triplicate blots run in parallel. *$p < 0.05$ (paired $t$ test). **e** Immunoblot analysis of phosphorylation events downstream of the BCR after stimulation of WT (black) or RhoG-deficient (red) B1-8[hi] B cells with a 3:1 ratio of bead-bound NIP-OVA for different time points. Plots to the right show protein phosphorylation levels relative to the amount of actin quantified by densitometry. Relative density is shown as the mean ± SEM of triplicate blots run in parallel. ***$p < 0.001$; *$p < 0.05$ (paired $t$ test). **f** Midplane confocal microscopy images of B1-8[hi] B cells in the process of phagocytosing (5 min. of incubation) or having completely phagocytosed (30 min.) 1 μm beads coated with NIP-OVA. Details of the phagocytic cups (5 min) and the phagosomes (30 min.), selected with yellow rectangles, are shown in the enlarged pictures. The B220 B cell marker is in blue (or red in the insets), beads in red, phospho-Igα, and phospho-Syk antibodies in green. Histogram overlays show the signal intensity in the three colors along the white lines drawn in the main images.

GC B cells such that, by day 5, two distinct populations of GC B cells are evident: Bcl-6[high]Blimp-1[low] and Bcl-6[low]Blimp-1[high] (Fig. 4d). Analyzing Bcl-6 and Blimp-1 expression according to the number of B cell divisions at day 3 showed that Bcl-6 expression reached a maximum at the second cell division, whereas Blimp-1 steadily increased up to the sixth cell division (Fig. 4e). The decline of Bcl-6 expression after 4 days of culture and the reduction of Bcl-6 expression in highly divided cells could reflect the repression exerted by Blimp-1 (Fig. 4d, e). These results indicate that B cell stimulation with a bead-bound antigen, which is taken up by phagocytosis, results in their differentiation to GC B cells in vitro with dynamic expression of Bcl-6 and Blimp-1, as shown previously in vivo[24,25].

The experiments above were carried out with B1-8[hi] B cells and OT-2 T cells purified by negative selection. To further rule out the participation of a third cell type that could be contaminating the B and T cell populations, the experiments with bead-bound versus soluble NIP-OVA antigen were repeated with flow cytometry-sorted follicular B (B220[+]CD23[+]CD43[−]CD11b[−]) and CD4[+] OT-2 T cells (Fig. S1a). In these conditions, B cells acquired GC markers (Fig. S1b) and differentiated into antibody-producing cells that secreted class-switched Igs (Fig. S1c), thus indicating that T and B cells are sufficient to generate the GC-like reaction. Overall, these data showed that incubation of naive B cells with a haptenated antigen immobilized onto 1 μm beads in the presence of antigen-specific helper CD4[+] T cells in vitro results in the generation of high-affinity antigen-specific antibodies of class-switched isotypes.

**Low-affinity B1-8[low] B cells also form GCs in vitro when stimulated with bead-bound antigen in the presence of OT-2 T cells**. The V region of B1-8[hi] knock-in mice bears a Trp to Leu substitution at position 33 that is frequently found in class-switched anti-NP antibodies and that increases affinity for NP by ~10-fold. To exclude the possibility that the generation of high-affinity antibodies in the in vitro GC system could be biased by the W33L mutation, we carried out similar experiments using B cells from another knock-in mouse line that maintains the germline Trp residue at position 33 (B1-8[low]). To determine if B1-8[low] B cells were also able to undergo Ig class switching in the in vitro GC system, we followed the generation of anti-NP antibodies by B1-8[low] B cells during 7 days of culture when stimulated with NIP-OVA-coated beads in the presence of OT2. The release of antibodies to the culture supernatant was quantified and compared to the effect of the general polyclonal B cell stimulus, LPS + IL4. The generation of anti-NP IgM upon

stimulation with NIP-OVA beads was slower than that produced with LPS + IL4 but reached an increased concentration by day 7 (400 ng/mL; Fig. S2a). Unlike the polyclonal LPS + IL4 stimulus, stimulation with beads coated with NIP-OVA antigen resulted in the generation of IgG1, IgA, IgG2a, and IgG3 starting from day 5 and reaching a maximum at day 7. The concentrations of anti-NP antibodies produced in the cultures were detectable, reaching an IgG1 concentration of ~15 ng/mL (Fig. S2a). As for B1-8[hi] + OT-2-cell cultures (Fig. 4), we followed the expression of GC and plasma cell markers by B1-8[low] cells upon stimulation of NIP-OVA beads in the presence of OT-2 cells. Formation of GC B cells according to the expression of Bcl6 and downregulation of CD38 reached a maximum at day 3 of culture (Fig. S2b) with NIP-OVA beads and was higher than that of B1-8[low] B cells stimulated with LPS + IL4. Expression of the GC markers GL7 and CD95 by B1-8[low] B cells was detected at day 2 and steadily increased until day 5 (Fig. S2c). B1-8[low] B cells stimulated with LPS + IL4 quickly upregulated GL7 and CD95, which remained high for the entire duration of the culture. The expression of CD138 and Blimp-1 by B1-8[low] B cells followed the peak of the GC phenotype (CD38-Bcl-6+) at day 3 and was maximal by day 5 (Fig. S2d), suggesting that beads coated with NIP-OVA were promoting the differentiation of GC B cells into plasma cells. Finally, the peak of GC B cells (Fig. S2b) was coincident with the peak of expression of the Tfh markers CXCR5 and PD1 by OT-2 T cells (Fig. S2d), suggesting that both processes were coordinated. Altogether, these data indicated that B1-8[low] B cells, stimulated with bead-bound antigen and OT-2 T cells, undergo antigen-driven antibody class switching and enter a GC-like reaction in vitro.

**B cells undergo somatic hypermutation and selection for higher affinity in the in vitro GC system**. In addition to the analysis of GC B cell markers by flow cytometry, we compared the transcriptional profile of the in vitro-generated GC B cells with that of sorted GC B cells generated in vivo after immunization with antigen. Both in vitro and in vivo GC B cells were purified by cell sorting using CD95 and GL7 expression as GC markers before RNA extraction. B cells stimulated in vitro with LPS and resting follicular B cells freshly purified from spleen of naive mice were used as controls. Data on mRNA expression for the four types of B cells are shown in Supplementary Data 1. From this dataset, we extracted information about the expression of genes that have been shown to be differentially upregulated in GC B cells over naive B cells (https://www.gsea-msigdb.org/gsea/msigdb/cards/GSE12366_GC_VS_NAIVE_BCELL_UP).

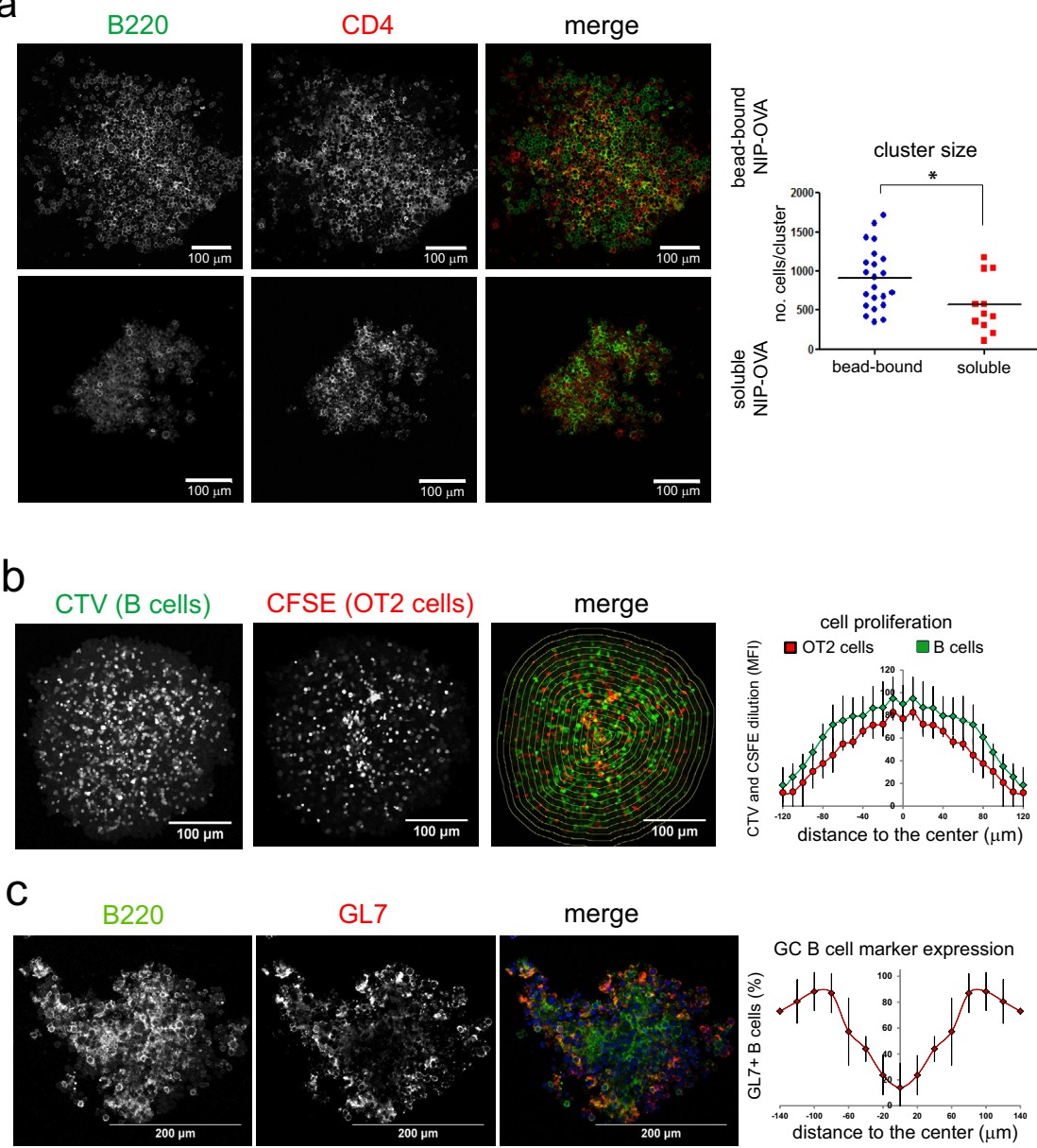

**Fig. 3 Phagocytic antigen induces the formation of large clusters of intermingled B and T cells. a** Confocal microscopy image of representative cell clusters generated after 4 days of co-culture of OT-2 T cells and B1-8[hi] B cells stimulated with NIP-OVA beads or with soluble NIP-OVA. B cells are stained with B220 in green; OT-2 T cells with CD4 in red. A quantification of the number of cells per cluster is shown in the plot to the right. Data represent the mean. *$p < 0.05$ (unpaired Student's $t$ test). **b** Confocal microscopy image of a cluster of B1-8[hi] B cells labeled with CTV and cultured for 7 days with CFSE-labeled OT-2 T cells and with beads coated with NIP-OVA. The intensity of CFSE and CTV staining was measured for all cells placed within the drawn concentric areas and represented in the plot to the right versus the distance to the center of the cluster. Data represent the mean ±S.D. for $n = 5$ clusters of similar size. **c** Confocal microscopy of a cluster generated by stimulation of B1-8[hi] WT B cells with beads coated with NIP-OVA for 7 days and stained with the B220 B cell marker (green) and the GL7 GC marker (red). GL7 intensity versus the distance to the center of the cluster was measured as in **b**. The line plot to the right represents the mean ±S.D. for $n = 5$ clusters of similar size. The experiments were repeated 5–10 times.

Expression of the GC gene set in GC B cells generated in vitro was much closer to that of GC B cells generated in vivo than to naive follicular B cells or LPS-activated B cells (Fig. 5a, Supplementary Data 1). These data strongly reinforce the idea that the in vitro system does recapitulate a GC.

The process of somatic hypermutation that leads to the selection of B cells with immunoglobulins of higher affinity for the inducing antigen is a hallmark of the GC reaction. To determine if there is somatic hypermutation in our in vitro system, we sequenced the $V_H$ region of B cell clones derived from B1-8[hi] B cells co-cultured for 7 days with OT-2 cells and with

either beads coated with NIP-OVA or soluble NIP-OVA. Sequencing by the Sanger method pointed to a frequency of mutations in the VH nucleotide sequence of $3.17 \times 10^{-4}$ for the stimulation with bead-bound NIP-OVA, almost 10-fold higher than the background frequency detected in non-stimulated naive B1-8[hi] B cells, and 50% higher than the frequency detected in B1-8[hi] B cells stimulated with soluble NIP-OVA (Fig. 5b). Bead-bound antigen led to a frequency of mutation at the protein level of $6 \times 10^{-4}$, which is 50% higher than the frequency generated by stimulation with soluble antigen and $10^5$-fold higher than the background mutation in naive B1-8[hi] B cells (Fig. 5b).

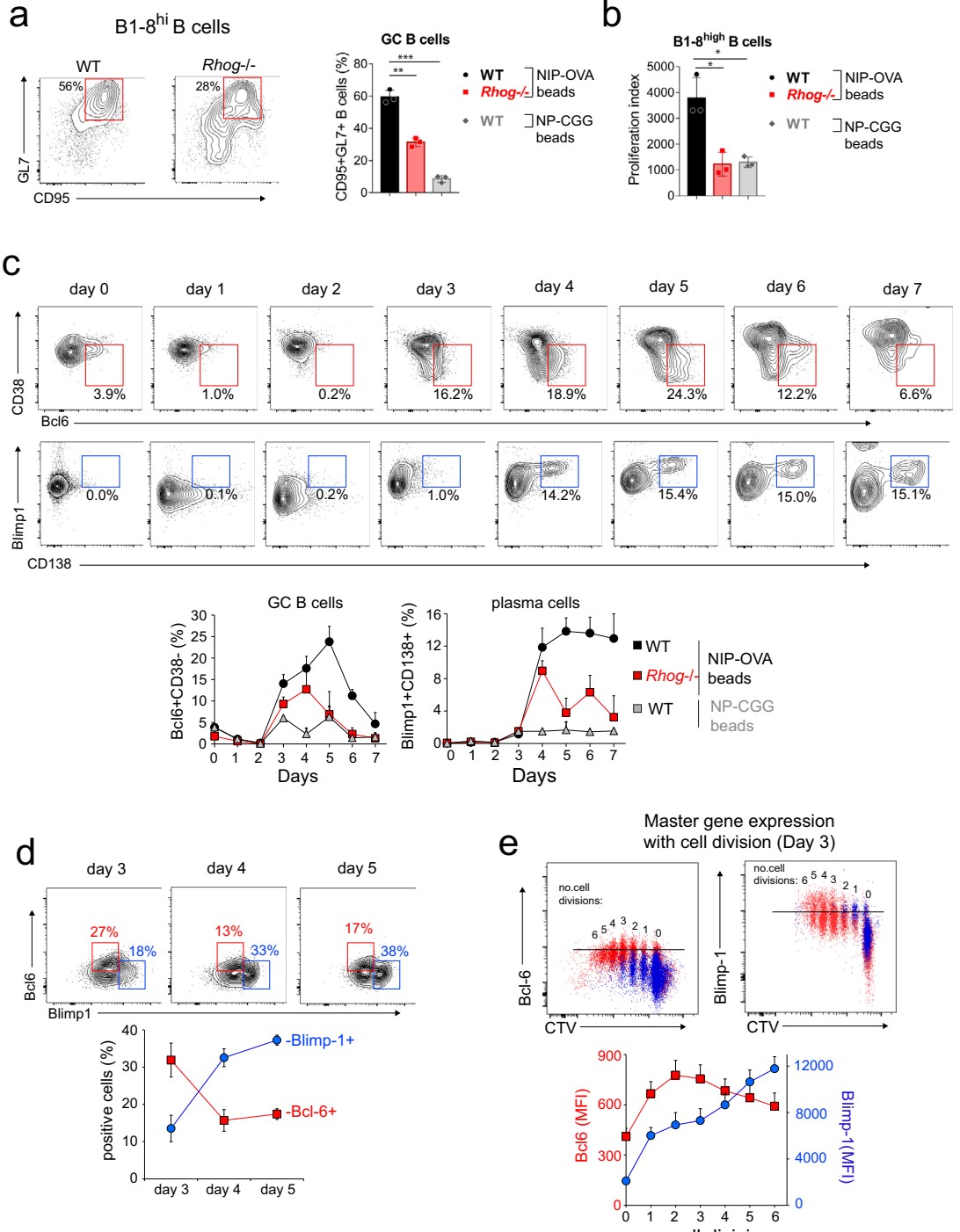

**Fig. 4 B cells differentiate in vitro into GC B cells upon their activation with phagocytic antigen and T cell help. a** Naive B cells from B1-8[hi] WT and *Rhog[−/−]* mice were pre-incubated with beads coated with NIP-OVA or NP-CGG, at a 3:1 bead/cell ratio, and co-cultured for 4 days with OT-2 T cells (1:1 B/T cell ratio). Flow cytometry contour plots to the left show the appearance of a double positive (CD95[+] GL7[+]) population in gated B220[+] B cells. The bar plot to the right represents mean ± S.D. (*n* = 3). **p < 0.01; ***p < 0.001 (unpaired Student's *t* test). **b** Proliferation of B cells from B1-8[hi] WT and *Rhog[−/−]* mice was calculated after 4 days of culture by CTV dilution as in Fig. 1. Data represent the mean ± S.D. (*n* = 3). *p < 0.05 (unpaired Student's *t* test). **c** Differentiation of naive B cells from B1-8[hi] WT and *Rhog[−/−]* mice into GC B cells and plasma cells was followed during 7 days of in vitro culture with beads coated with either NIP-OVA or NP-CGG and OT-2 CD4[+] T cells. The percentages of GC B cells were calculated according to CD38 downregulation and expression of intracellular Bcl-6 by gated B220[+] B cells. The percentages of plasma cells were calculated according to the expression of CD138 and intracellular Blimp-1 by gated B220[+] B cells. Representative plots for WT cells are shown. Line plots below represent mean ± SD (*n* = 3). **d** Expression of Bcl-6 and Blimp-1 by GC B cells was assessed by intracellular staining of B220[+] B cells in B1-8[hi]/OT-2 CD4[+] T cell cultures stimulated with NIP-OVA beads. The line plots below represent mean ± SD values (*n* = 3). **e** Expression of Bcl-6 and Blimp-1 as a function of the number of cell divisions by naive B cells from B1-8[hi] WT mice stimulated 4 days in vitro with beads coated with either NIP-OVA (red dots) or NP-CGG (blue dots) and OT-2 T CD4[+] T cells. The number of cell divisions was assessed by CTV dilution.

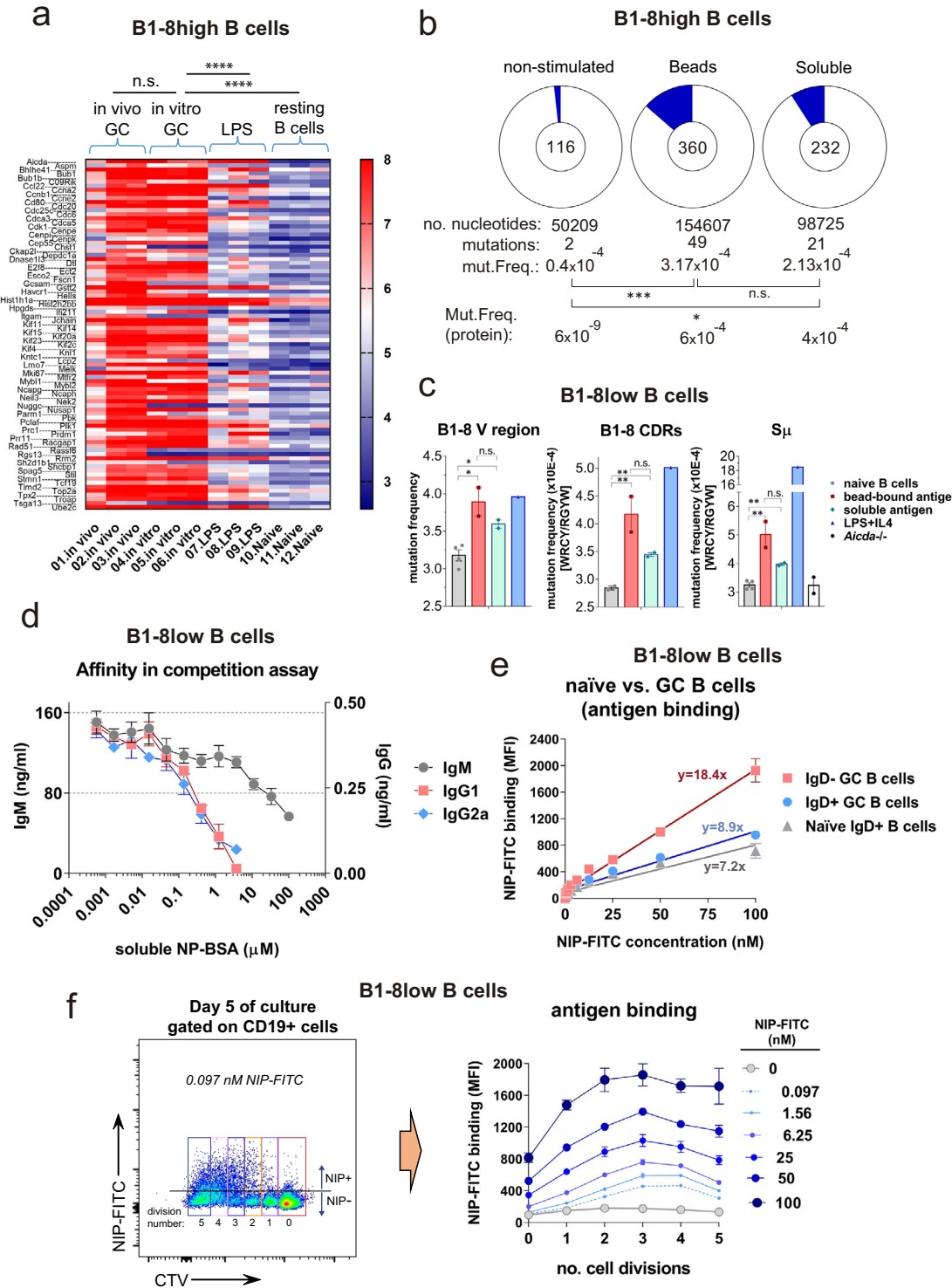

Interestingly, the amino acid-changing mutations clustered at the three complementary loops of the $V_H$ sequence or in their close proximity (Fig. S3). Therefore, these data show that B1-8$^{hi}$ B cells undergo somatic hypermutation in the in vitro antigen-driven GC reaction.

To determine if B1-8$^{low}$ B cells also undergo somatic hypermutation in vitro, we carried out a deep-sequencing analysis of the 5-day GC B cell population by PCR-seq. We analyzed the mutation frequency of the Switch (S) region first, which is not associated with affinity maturation, and it has been described to occur both in vivo and in vitro under LPS stimulation.

We concentrated the analysis of mutation frequency in mutational WRCY/RGYW hotspots in the non-coding Sμ region of the IgH locus[26]. We used both naive follicular B cells and B cells from activation-induced cytidine deaminase (AID) knockout mice to establish background levels of mutation. The analysis showed a mutation frequency within AID hotspots in the Sμ region of $5 \times 10^{-4}$ for bead-bound antigen stimulation, which was significantly higher than that of naive B cells, and higher than stimulation with soluble antigen (Fig. 5c). B cells stimulated with LPS + IL4 had a threefold higher mutation rate in the Sμ region when compared with bead-bound antigen in the presence of

**Fig. 5 In vitro GC B cells undergo somatic hypermutation and affinity maturation and express a transcriptional fingerprint of in vivo GC. a** Heatmap plot showing mRNA expression of a selected GC gene set after microarray analysis of sorted: **1)** CD19 + CD95 + GL7 + GC B cells from spleens of B1-8$^{hi}$ mice 7 days after immunization with NP-CGG in alum; **2)** CD19 + CD95 + GL7 + B cells from B1-8$^{hi}$ mice stimulated in vitro with bead-bound NIP-OVA beads together with OT-2 T cells for 3 days; **3)** CD19 + B cells from B1-8$^{hi}$ mice stimulated in vitro for 3 days with LPS + IL4 (LPS); **4)** CD19 + CD23 + CD21$^{low}$ follicular B cells freshly isolated from B1-8$^{hi}$ mouse spleens (resting B cells). Each column represents data from a single mouse or B cell culture. **b** Generation of somatic mutations in IgH V genes in conditions of GC formation in vitro. Number and frequency of nucleotide mutations by Sanger sequencing in the B1-8V$_H$ sequence of sorted B cells from B1-8$^{hi}$ mice stimulated with a 3:1 bead/cell ratio of beads coated with NIP-OVA or with 100 ng/ml of soluble NIP-OVA and co-cultured for 7 days with OT-2 T cells. Frequency of non-silent protein mutations is also indicated. Blue segments in the pie charts are proportional to the number of sequences carrying mutations. The total number of independent sequences analyzed is indicated in the center of each chart. The calculated mutation frequency per base pair is indicated underneath. **c** Mutation analysis by NGS of Sμ and B1-8V$_H$ regions in sorted B cells from B1-8$^{low}$/OT-2 CD4$^+$ T cell cultures stimulated with NIP-OVA beads (bead-bound) or soluble NIP-OVA (soluble antigen) for 5 days. naive B1-8$^{low}$ cells at day 0 and after 3 days of stimulation with LPS plus IL4 were also analyzed. B cells from Aicda$^{-/-}$ mice were used as negative control for technical background (empty bar). Mutation frequency at G/C lying in WR<u>C</u>Y/R<u>G</u>YW AID hotspots is shown. **d** Estimation of antibody affinity by competition assay. A 7-day culture supernatant of B1-8$^{low}$/OT-2 CD4$^+$ T-cell cultures stimulated with NIP-OVA beads was incubated with the indicated concentrations of soluble NP-BSA in competition for binding to NP-BSA immobilized on an ELISA plate. The actual concentrations of Igs was determined using standard curves as described in Methods. Data represent mean ± s.e.m. of cultures in triplicate. **e** Evidence of affinity maturation shown by comparing the binding of a range of NIP-FITC concentrations to freshly isolated naive B1-8$^{low}$ B cells, to IgD+ B cells, and to IgD− B cells after 5 days in the B1-8$^{low}$/OT-2 in vitro GC system. Binding is shown as the mean fluorescence intensity (MFI) determined by flow cytometry. Data represent mean ± s.e.m. of cultures in triplicate. Values were adjusted to lineal regressions (R$^2$ > 0.9). The slopes of the lineal equations are indicated. **f** Evidence of affinity maturation gathered from NIP-FITC antigen-binding data to B1-8$^{low}$ B cells labeled with CTV and according to their number of cell divisions. After 5 days of culture in the B1-8$^{low}$/OT-2 in vitro GC system, B cells were incubated with the indicated concentrations of NIP-FITC antigen. The pseudocolor plot on the left illustrates how NIP-FITC at very low concentration (0.097 nM) binds stronger to divided cells than to non-divided ones. Line plots to the right shows binding data for all concentrations. Data represent mean ± s.e.m. of cultures in triplicate.

OT-2, but not on the B1-8$^{low}$ V$_H$ region or the antigen-binding CDR loops (Fig. 5c). Interestingly, and unlike previous reports, we do detect somatic mutations in the V$_H$ region using our in vitro GC system but also LPS + IL4 stimulation[27]. Probably, the differences with previous findings are due to our use of deep-sequencing technologies in the analysis of AID hotspots.

We next interrogated if, in our in vitro GC system, there is not only somatic hypermutation, but also selection for B cells expressing BCRs of higher affinity. We first carried out a competition ELISA assay to estimate the average affinity for both IgM and class-switched antibodies generated in the in vitro antigen-driven GC reaction. Soluble NP-BSA was added at different concentrations to compete with immobilized NP hapten for binding to antibodies contained in the culture supernatants. We found that the concentration of soluble NP-BSA that inhibited anti-NP IgM binding by 50% was ~20 μM, whereas the concentration needed to inhibit 50% of anti-NP IgG1 binding was ~300 nM and ~200 nM for anti-NP IgG2a (Fig. 5d). These results suggest that stimulation with bead-bound antigen of B1-8$^{low}$ B cells in the in vitro GC system results in immunoglobulin class switching that is accompanied by increased affinity for antigen.

To visualize the selection for high-affinity antigen-specific B cells in a more direct manner, we carried out a similar experiment using CTV-labeled B1-8$^{low}$ B cells. This allows to determine the capacity to bind NIP hapten antigen of B cells that had gone through the GC process compared to naive B1-8$^{low}$ B cells. Binding to NIP-FITC in response to increasing concentrations of antigen was measured by flow cytometry and data was plotted and fitted to a linear function. The slope for NIP-FITC binding of B1-8$^{low}$ B cells that had undergone Ig class switching (IgD−) after 5 days in culture was higher than that of IgD+ GC B cells and slightly higher than that of naive B1-8$^{low}$ B cells at day 0, i.e. before being subjected to the GC process (Fig. 5e). These results suggest that B cells with higher affinity for NIP antigen were selected during the GC process in vitro. Furthermore, B cells that had undergone 1–5 rounds of cell divisions, calculated according to CTV dilution (Fig. 5f, left), were able to bind NIP-FITC at concentrations of antigen as low as 0.097 nM, whereas non-divided cells did not show binding higher than the background

NIP-FITC fluorescence levels (Fig. 5f, right). Altogether, the results in Fig. 5d–f show that there is a selection in the in vitro GC system for B cells expressing and secreting Igs of highest affinities.

To determine if the B cells generated in the in vitro GC system can functionally integrate within a GC response in vivo, we carried out adoptive transfer experiments in which wild-type C57BL/6 mice bearing the CD45.2 allele were intravenously inoculated with B1-8$^{low}$ B cells bearing the CD45.1 allele that had been stimulated for 5 days in the in vitro system. We analyzed the response of those cells to a second stimulation in vivo compared to that of naive CD45.1 + B cells that had not been stimulated in vitro. Those results showed that B cells selected in the in vitro GC system generate more NIP + Ig class-switched (IgD−) cells upon immunization in vivo and more NIP + IgG1+ cells than naive adoptively transferred (CD45.1+) or naive endogenous (CD45.2+) B cells (Fig. S4).

**Cognate and not bystander B cells can make high-affinity class-switched antibodies in the in vitro GC system.** The in vitro antigen-driven GC system described here offers the possibility of studying T-B cell cooperation at a level of detail that cannot be achieved in vivo. As an example of the possible uses of the method, we aimed to investigate if T cells confer help only to cognate antigen-presenting B cells or can also provide help to nearby bystander B cells that are also stimulated by antigen but cannot present it to T cells. The underlying question here is: do bystander autoreactive B cells undergo Ig class switch and affinity maturation with the help of T cells that are responding to a foreign antigen? To model this, we used B1-8$^{low}$ B cells, OT-2 T cells, and B cells from another knock-in mouse line expressing a BCR reactive to hen-egg lysozyme (SWHEL). In the spleen, 35% of B cells express the SWHEL BCR and are able to bind biotinylated HEL (Fig. S5a). In one setting, B1-8$^{low}$ B cells were stimulated with bead-immobilized NIP-OVA and therefore, needed to phagocytose antigen to present OVA peptides to OT-2 (setting [A], Fig. 6a). SWHEL B cells in this setting are bystanders because, although stimulated through their BCR with soluble antigen (HEL), they could not present cognate antigen to OT-2 T cells. In another setting, SWHEL B cells were stimulated with

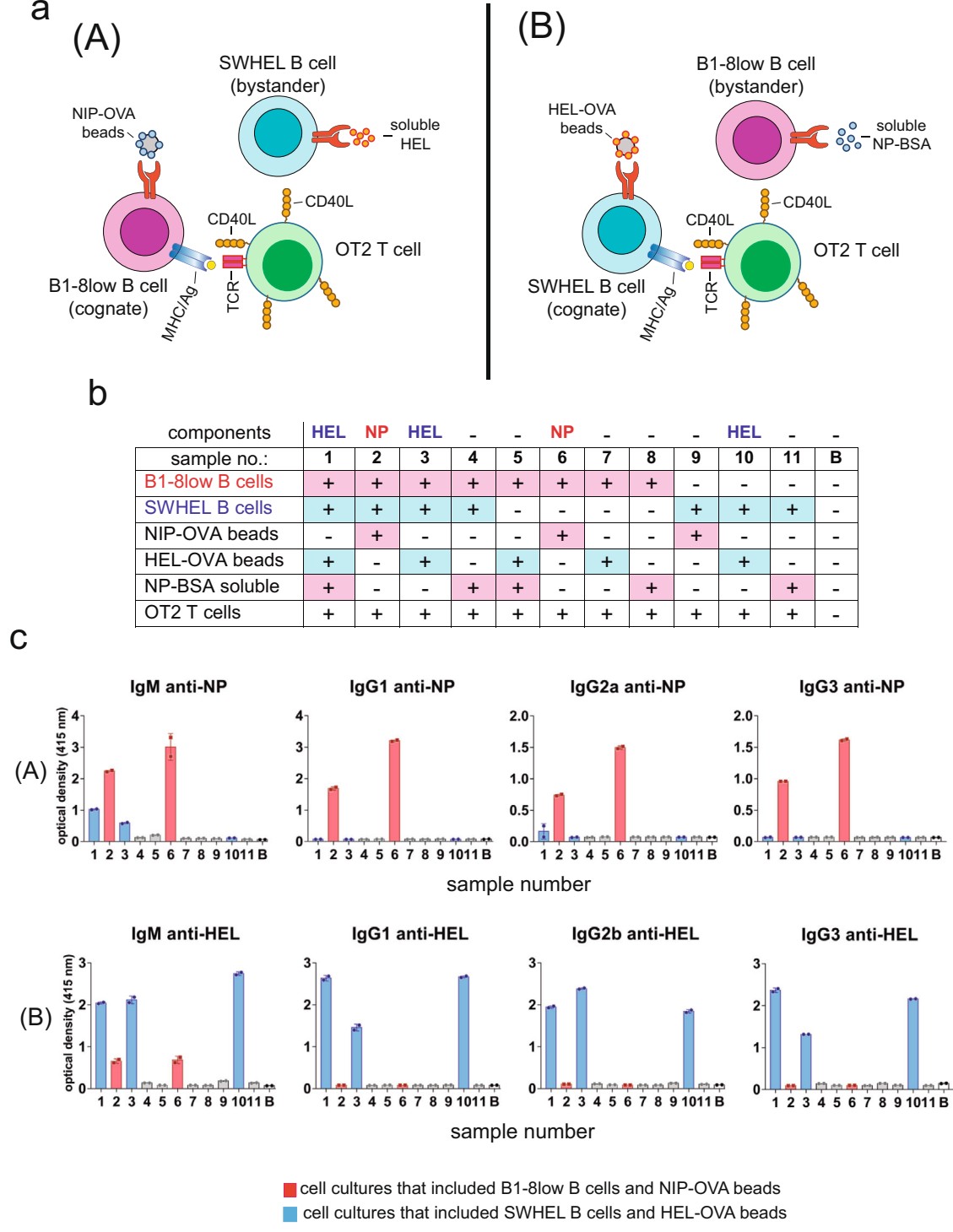

**Fig. 6 Cognate B cells but not bystander B cells make antigen-specific class-switched immunoglobulins in the in vitro GC system. a** Cartoon summarizing the two experimental settings. In setting (A), B1-8$^{low}$ are the cognate B cells that take up NP-OVA antigen by phagocytosis and present OVA peptide to OT-2 CD4 + T cells. SWHEL are the bystander B cells that are stimulated with soluble HEL. In setting (B), SWHEL are the cognate B cells that take up HEL + OVA antigen from coated beads and present OVA peptide to OT-2 CD4 + T cells. B1-8$^{low}$ are the bystander B cells that are stimulated with soluble NP-BSA. **b** Summary of the 11 different cell co-cultures organized according to settings (A) and (B), containing different mixtures of cells and antigens. B, blank culture medium. All culture supernatants were harvested after 7 days. **c** Detection of IgM and class-switched Igs specific for HEL and for NP in the 11 culture mixes plus blank, described in **b**. Data represent the mean ± S.D. ($n = 3$).

beads coated with HEL and OVA, and therefore could present cognate antigen to OT-2 cells, whereas B1-8$^{low}$ B cells stimulated with soluble NP-BSA were the bystander B cells (setting [B], Fig. 6a). Using the two settings, we designed a panel of

experimental points in which we included different controls, as indicated in Fig. 6b. After 7 days of culture, the supernatants were tested for the presence of IgM and class-switched antibodies specific for NP and for HEL. Only those cultures in which the

cognate B cells were stimulated with bead-bound antigen generated IgM and IgGs (samples 1, 2, 3, 6, and 10, Fig. 6c). Stimulation of B1-8$^{low}$ B cells with NIP-OVA resulted in the generation of IgM, IgG1, IgG2, and IgG3 in the presence or absence of bystander SWHEL B cells (samples 2 and 6). Interestingly, stimulation of cognate SWHEL B cells with HEL-OVA beads promoted bystander B1-8$^{low}$ B cells to produce anti-NP IgM, but not IgGs (samples 1 and 3). Likewise, stimulation of cognate B1-8$^{low}$ B cells with NIP-OVA beads promoted bystander SWHEL B cells to produce anti-HEL IgM but not IgGs (sample 2), although this result was obscured by the result in sample 6 in which B1-8$^{low}$ B cells, in the absence of SWHEL B cells, produced anti-HEL IgM after stimulation with NIP-OVA beads. This result might be due to likely contamination of the preparation of NIP-OVA (egg ovalbumin) with egg lysozyme (HEL) and to the fact that B1-8$^{low}$ mice also have a polyclonal B cell repertoire.

The differences between cognate and bystander B cells in their capacity to produce class-switched antibodies were not due to the soluble format of the antigen-specific for the bystander cells. If in setting [B] (Fig. 6a) soluble NP-BSA is replaced by bead-bound antigen, the bystander B1-8$^{low}$ B cells were still deficient in the production of class-switched IgG1 and IgG2a antibodies (Fig. S5b) and in the generation of IgG1+ class-switched B cells (Fig. S5c), compared to cognate SWHEL cells.

### Early formation of conjugates with Tfh cells by cognate B cells.
To understand the reasons why bystander B cells make IgM but not class-switched antibodies, we took advantage of the cell and stimulation conditions used in sample 1 above (Fig. 6b, c), in which cognate SWHEL B cells are stimulated with HEL-OVA beads in the presence of OT-2 T cells and bystander B1-8$^{low}$ B cells are stimulated with soluble NIP-BSA antigen. We followed the secretion of IgM and IgG1 of low and high affinity during 7 days of culture, detecting IgM of low-affinity against HEL and NP starting at days 2–3 and reaching a maximum at day 6 (Fig. 7a). The release of anti-HEL IgG1 was detected first at day 5 and reached a maximum at days 6–7. No anti-NP IgG1 was detected at any time point (Fig. 7a). These data further demonstrate that there is no antibody class-switching by the bystander B1-8$^{low}$ B cells. The secretion of anti-HEL IgG1 was mirrored by the detection of up to 30% of B cells expressing cell surface IgG1 and a loss of membrane IgM expression (Fig. 7b). Bystander B1-8$^{low}$ cells retained surface expression of IgM and did not express membrane IgG1.

The release of anti-HEL and anti-NP IgM to the culture supernatant starting at days 2-3 was accompanied by the detection of Blimp1 + CD138 + plasma cells of cognate and bystander B cell origin (Fig. 7c). However, the percentage of cognate SWHEL plasma cells increased at day 5 and continued increasing at days 6 and 7, whereas the percentage of bystander B1-8$^{low}$ plasma cells steadily declined after day 4 (Fig. 7c). Both cognate and bystander B cells expressed GL7 and CD95 beginning at days 1–2, although the percentage of these was higher within cognate B cells than within bystanders (Fig. 7d). In addition, the percentage of GC B cells within the cognate B-cell population was sustained from days 2 to 7, whereas the percentage within the bystander B cell population declined after day 3.

In the 3-cell culture system, the differentiation of OT2 CD4 + T cells into CXCR5 + PD1 + Tfh cells reached a maximum of 70% of the T cell population at day 3 (Fig. 7e). Those CD4 + T cells were bona fide Tfh because they also expressed the master regulator Bcl6 (Fig. 7f). We followed the formation of T-B cell conjugates by flow cytometry according to the gating strategy indicated in Fig. S6. By analyzing the expression of Tfh markers within the T-B conjugates we found that, in spite of them being a

smaller population at day 1 (Fig. 7e), most T-B conjugates with cognate B cells were already formed with Tfh rather than with naive CD4 + T cells (Fig. 7g). The preferential formation of cognate B cell conjugates with Tfh was also detected at day 2, when the Tfh population remained a minority. Bystander B cells formed less conjugates with Tfh (Fig. 7g) than cognate ones at every time point. Interestingly, unlike for cognate B cells, the maximum of conjugate formation between bystander B cells and Tfh was seen at day 3, at the peak of Tfh differentiation (Fig. 7g, e). Overall, the most striking difference between cognate and bystander B cells was the early (days 1–2) preferential formation of conjugates with Tfh cells. Altogether, the 3-cell co-culture results indicate that bystander B cells can be induced to produce IgM in the vicinity of GC-forming B cells, but they will not receive T cell help to undergo Ig class switching. Also, T cells are capable of forming conjugates repeatedly, first as naive T cells and as Tfh cells after first stimulation (day 1).

### Bystander B cells divide poorly and acquire an early memory phenotype.
To investigate the rate of cell division by cognate and bystander B cells simultaneously, we labeled the cognate B SWHEL B cells with CTV and the bystander B1-8$^{low}$ B cells with Cell Trace Far Red (CTFR) to measure cell division according to dye dilution. At day 3, most cognate B cells had undergone 3-cell divisions, whereas most bystander B cells had divided only once (Fig. 8a). At days 5 and 7, the population of cognate B cells had undergone 3–7 divisions, while the number of cell divisions of bystander B cells peaked at day 3 and then decreased (Fig. 8a and Fig. S7a). These results suggest that, unlike cognate B cells, the proliferation of bystander B cells is halted after day 3. The total number of cognate and bystander B cells decreases by ~50% from days 0 to 7 (Fig. 8b). However, there were no significant differences in the number of live cognate and bystander B cells throughout the experiment. Therefore, it does not seem that the early halt in cognate cell division at day 3 is due to overall increased mortality.

The marker CD38 distinguishes naive B cells (CD38+) from GC B cells (CD38$^{low}$) and is re-expressed on memory B cells which become again CD38$^{high}$ [28]. We used CTV or CTFR dilution together with CD38 expression to distinguish naive, GC, and memory B cells. Divided B cells that downregulate CD38 will be GC B cells, whereas divided B cells that are CD38$^{high}$ will be memory B cells. We found that, by day 3, 50% of the cognate and bystander B cells that had undergone 3-cell divisions had also downregulated CD38, suggesting they were GC B cells (Fig. S7b). Furthermore, most of the cognate and bystander B cells that had divided 5 times were CD38$^{low}$ (Fig. S7b). By day 5, this percentage of CD38$^{low}$-divided B cells diminished considerably within the cognate B cell population but had virtually disappeared within the bystander cells (Fig. 8c). These results suggest that poorly divided bystander B cells acquire an early phenotype of memory B cells.

To confirm this hypothesis, we analyzed the expression of CD73, another marker of B cell memory[29] at day 3. We found that divided bystander B cells expressed CD73, in addition to high levels of CD38, while cognate B cells expressed less CD73, confirming the idea of an early conversion of bystander B1-8$^{low}$ IgM+ GC B cells into memory B cells[30] (Fig. 8d). Although, CD73 and other markers are expressed by activated B cells, previous works in mice immunized with haptenated antigens or infected with influenza virus or Plasmodium sp. have helped to propose different types of memory B cells according to the expression of CD73, CD80, and PDL2 among others[31]. Memory B cells expressing CD73, but not PDL2 or CD80, define a stem-like memory subset with low Ig mutational content and high

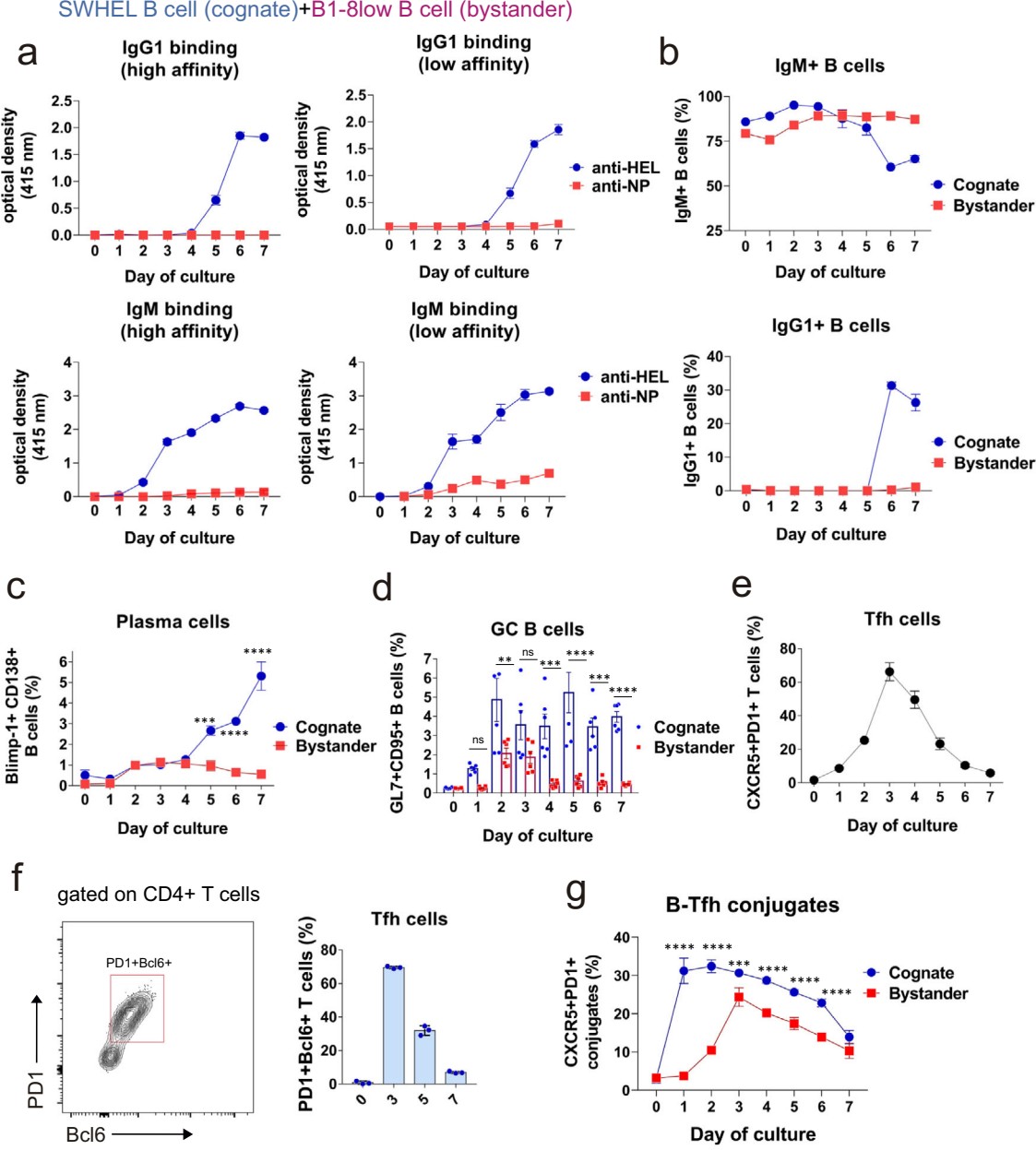

**Fig. 7 Cognate B cells are more efficient than bystander B cells at making GC B cells and plasma cells in a sustained manner. a** Following the setting (B) (Fig. 6a) in which B cells from SWHEL mice were stimulated with beads coated with HEL + OVA and B1-8$^{low}$ B cells were stimulated with soluble NP-BSA, in the presence of OT-2 CD4 + T cells, culture supernatants were harvested along 7 days and the concentration of high- and low-affinity IgM and IgG1 was determined by ELISA. Data represent the mean ± S.D. ($n = 3$). **b** The cell pellets of the three-cell type cultures in **a** were examined by flow cytometry according to the expression of membrane IgM and IgG1. Cognate and bystander B cells were distinguished according to the expression of the CD45.2 and CD45.1 markers. Data represent the mean ± S.D. ($n = 3$). **c** Following the general setting (B) used in Fig. 6a, the presence of plasma cells in the cell three-cell type cultures was analyzed according to the surface expression of CD138 and intracellular Blimp1 in CD45.2+ and CD45.1 + B cells. Data represent the mean ± S.D. ($n = 3$). ***$p < 0.001$; ****$p < 0.0001$ (two-way ANOVA test). **d** Likewise, the presence of GC cells in the cell three-cell type cultures was analyzed according to the surface expression of GL7 and CD95 in CD45.2+ and CD45.1 + B cells. Data represent the mean ± S.D. ($n = 3$). **$p < 0.01$; ***$p < 0.001$; ****$p < 0.0001$; ns, not significant (two-way ANOVA test). **e** In parallel to plasma cell and GC B-cell phenotyping, the presence of Tfh cells within the CD4 + OT-2 T cell population was assessed by CXCR5 and PD1 staining. Data represent the mean ± S.D. ($n = 3$). **f** In a similar setting the presence differentiation of OT-2 T cells into Tfh was assessed by intracellular staining with Bcl6. The contour plot on the left illustrates how PD1 + Bcl6 + CD4 T cells are seen in the flow cytometer. The bar plot to the right shows the mean ± s.e.m. ($n = 3$) of PD1 + Bcl6+ cells in percentage within total CD4 T cells in function of the day of culture. **g** Frequency of cognate and bystander B cell conjugates with CD4 + OT-2 T cells that express the CXCR5 and PD1 Tfh markers at different days of co-culturing. Data represent the mean ± S.D. ($n = 3$). **$p < 0.01$; ****$p < 0.0001$ (two-way ANOVA test).

plasticity. Other subsets of memory B cells do express CD80 and PDL2 and contain B cells that have undergone Ig class switching, more somatic hypermutation, and have higher affinity for antigen. Those CD80 + PDL2 + differentiated memory B cells

have a greater propensity for differentiation into plasma cells in the secondary response[31]. Bystander B1-8$^{low}$ B cells were deficient in the expression of CD80 (but not of CD73 and CD38) in the IgD- class-switched population, compared to cognate SWHEL B

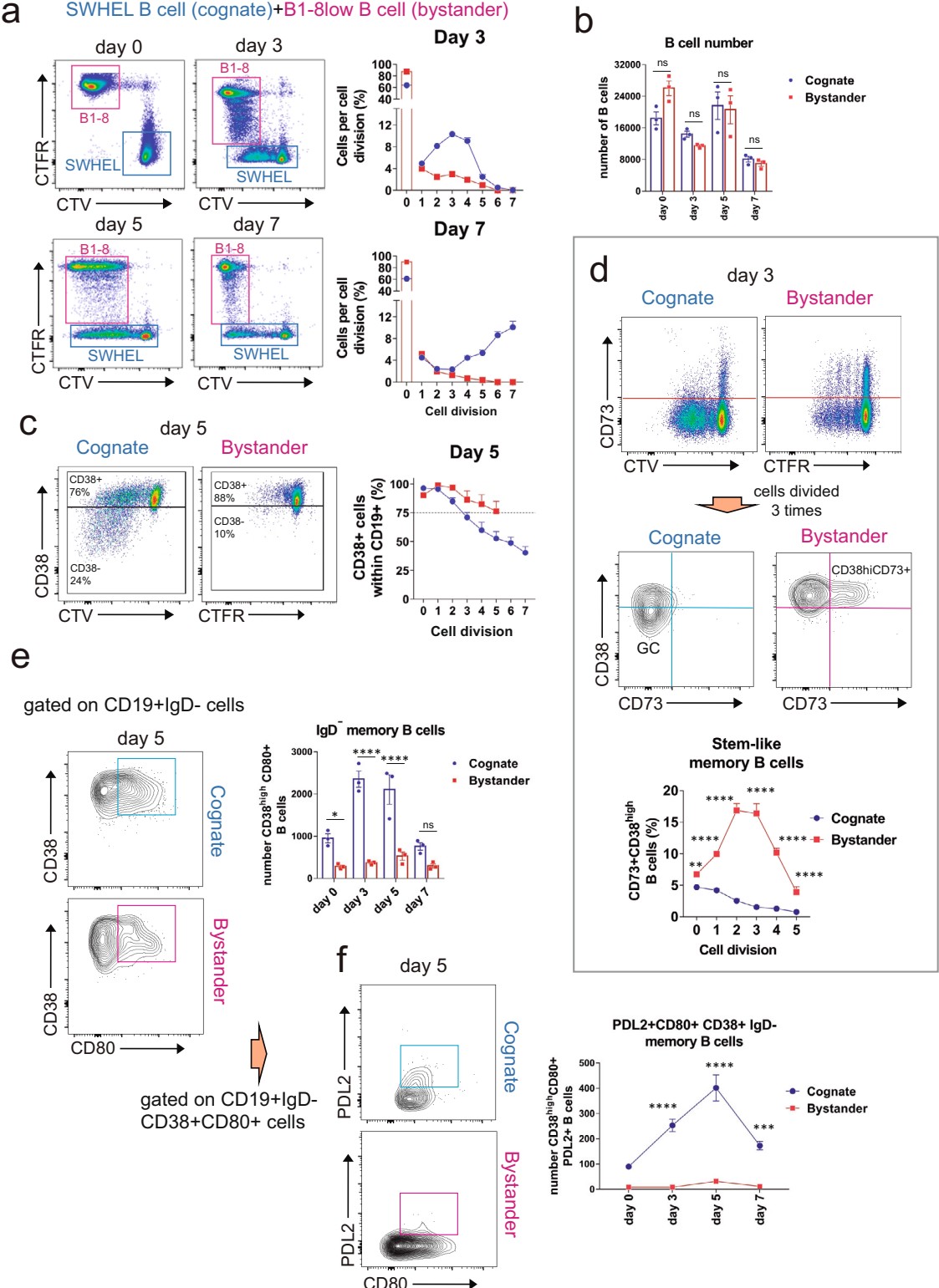

cells (Fig. 8e). Bystander memory B cells were also deficient in the expression of the PDL2 marker (Fig. 8f). Altogether, these results suggest that bystander B cells and cognate B cells have a distinct capacity to differentiate into memory B cell subsets, i.e., whereas bystander B cells differentiate early on into a stem-like IgD+ memory subset, cognate B cells differentiate later into an Ig class-switched memory subset that is more oriented towards differentiation into plasma cells after a second challenge.

## Discussion

In this study, we describe a system to recreate a GC in vitro based on the use of just two cell types, namely, B cells and CD4 + T cells, and on the administration of antigen to B cells associated to a particle that needs to be phagocytosed by a BCR-mediated mechanism[11]. Unlike previously described methods[11], the phagocytosis-based method recreates an antigen-specific GC reaction and can therefore be used to study T-B cell interactions.

**Fig. 8 Short proliferative span and early conversion into B cell memory of bystander B cells. a** Following the general setting (B) of Fig. 6a and Fig. 7, proliferation of cognate and bystander B cells was simultaneously followed according to the dilution of the Cell Tracer Violet (cognate, SWHEL) and the Cell Tracer Far Red (bystander, B1-8low). Data represent the mean ± S.D. (n = 3). **b** The number of live cognate and bystander B cells was calculated at the indicated days of co-culturing using counting beads. Data represent the mean ± S.D. (n = 3). ns, not significant (p > 0.05, two-way ANOVA test). **c** Cognate and bystander B cells were analyzed for the expression of the CD38 marker according to the number of cell divisions and the indicated days of co-culturing. Data represent the mean ± S.D. (n = 3). **d** Cognate and bystander B cells were analyzed for the expression of the CD73 marker according to the number of cell divisions at day 3 of co-culturing. Cells divided 3 times, according to CTV or CTFR dilution, were reanalyzed for the expression of CD38 and CD73. Cognate SWHEL B cells show the presence of a CD38high (memory) and a CD38low (GC) populations that were CD73−. Bystander B1-8low B cells have a population of CD38highCD73 + memory cells. Quantification of B cells with that memory phenotype is shown in the line plot below. Data represent the mean ± S.D. (n = 3). **p < 0.01; ****p < 0.0001 (two-way ANOVA test).

We show that B cells that have phagocytosed 1 μm-large inert particles coated with specific antigen can efficiently present peptide-antigen derived from these particles to CD4+ T cells in vitro. In exchange, T cells proliferate and express markers of Tfh differentiation. In turn, Tfh cells provide help to the antigen-presenting B cells, promote Ig class switching, and favor their differentiation into GC B cells and somatic hypermutation, two key features of a mature GC-derived humoral response. Indeed, using this simple two cell-system, we detect the production of high-affinity antigen-specific antibodies of all isotypes. Furthermore, the in vitro-generated GC B cells can be inoculated into mice in vivo and reengaged into subsequent humoral secondary responses, thus demonstrating that those cells generated in vitro are not rendered unresponsive. Therefore, we have devised a method to recreate antigen-specific GCs in vitro.

One of the keys to the success of the GC process is the acquisition of antigen through a BCR-dependent phagocytic process. This approach is linked to stronger and more sustained BCR signaling that cannot be achieved by soluble antigen. During phagocytosis, BCR signaling initiates in the phagocytic cup and persists once the particle has been completely phagocytosed. The phagocytic process is mediated in part by the GTPase RhoG[8], which has been previously involved in the phagocytosis of apoptotic bodies by macrophages, and in both type I and type II phagocytosis[13,16]. Although the defect in RhoG does not lead to a blockade of B cell phagocytosis and GC formation in vitro, perhaps because of a possible redundancy with other Rho GTPases[16], we previously showed that RhoG-deficient B cells are a valuable tool to determine the role of antigen phagocytosis by B cells in the humoral response in vivo[8]. Although it has been previously shown that B1 cells and, to a lesser extent, follicular B2 cells can phagocytose bacteria[32–34], the prevalent view is that B cells acquire antigen from immune complexes retained at the surface of FDCs or macrophages[3,5]. Although our data do not oppose this view, they highlight the relevance of the phagocytic process for GC differentiation. Indeed, it has been shown that antigen is captured by binding to the BCR in a process that also physically extracts membrane components from the cell that presents antigen to the B cell[35]. This phenomenon is very much reminiscent of the trogocytosis process by which T cells acquire MHC and membrane fragments from APC and that we previously characterized as a phagocytic process mechanistically mediated by RhoG[17]. Therefore, it could be that the mechanism of acquisition of antigen by B cells from follicular dendritic cells (FDCs) and macrophages displaying immune complexes is indeed phagocytic.

Still, what is the relevance of antigen phagocytosis by B cells in the GC response? It has been shown that the endocytosed BCR continues signaling from intracellular compartments, which is important to sustain the activation of the PI3K-Akt pathway[36]. We show here that phagocytosis promoted by the BCR results in a stronger and more sustained phosphorylation of the BCR and downstream targets (including Akt) than a soluble stimulus, presumably internalized by B cells via an endocytic mechanism.

Phagocytosis can also be a mechanism for selection of B cells bearing BCRs with higher affinity for antigen. High affinity for antigen correlates with enhanced CD4+ T cell activation[37] and mechanical forces are used to discriminate between antigen affinities by B cells[38]. Interestingly, antigen uptake by B cells from artificial membranes require myosin II, which is known to be required for phagocytic cup squeezing[39,40]. Hence, the energetic requirements for phagocytosis may make this mechanism of antigen uptake by B cells a discriminatory sensor of BCR affinity. Indeed, we show here that in the in vitro GC system there is selection for B cells bearing membrane Igs of higher affinities.

Compared to in vivo studies, an advantage of an in vitro method for recreation of GC in an antigen-dependent fashion is the capacity to study T-B cell cooperation in detail during the GC response in a timed manner. As an example, we have used the method to ask if, within an ongoing GC reaction with T cells providing help to cognate B cells, bystander B cells that are not presenting antigen to T cells will respond to T cell help. In our model system with a cognate B cell type, a bystander B cell type, and an antigen-specific cognate T-cell type, we observe that bystander B cells do not undergo Ig class-switching, but do proliferate, express GC B cell markers and release antigen-specific IgM. The limited proliferation of bystander B cells may explain the absence of class switching in this cell population[41]. This finding correlates with reduced formation of T-B conjugates by bystander B cells and increased and early expression of CD38 and CD73 markers of memory within divided bystander B cells. Different models of memory B cell versus plasma cell development have been proposed (reviewed in ref. [42]). Our data support an Instructive Fate Model in which B cells that receive sufficient BCR signals but receive weak T cell help become committed to a memory B cell fate. The expression of CD73 and low or negative expression of CD80 and PDL2 suggest that the early differentiation of bystander B cells into B cell memory is indeed a differentiation towards a subset of stem-like memory B cells that have not undergone Ig class-switching[31]. The outstanding question is: what weakens T cell help to bystander B cells? Our working hypothesis is that T cell help is provided to B cells that are presenting their cognate antigen in a vectorial polarized manner towards the immunological synapse and not in any other location of the T cell's plasma membrane. Verifying this hypothesis will require high-resolution optical microscopy using the antigen-specific GC in vitro system described here, beyond the scope of the present paper.

In summary, our study shows a method to recreate antigen-specific GC responses in vitro that can be used to mechanistically study T cell: B cell interaction requirements for GC responses in detail. As an example of its possible uses, our in vitro GC system shows that T cells might control the antigen specificity of B cells that undergo Ig class switching by restricting their help to cognate B cells that present antigen. This mechanism might be fundamental to avoid help to bystander B cells expressing autoreactive IgM specificities.

## Methods

**Mice and cells**. B1-8[hi] knock-in mice bear a pre-rearranged V region specific for the hapten 4-hydroxy-3-nitrophenylacetyl (NP)[19]. These mice were crossed with *Rhog*[−/−] mice[43] to generate B1-8[hi] B cells deficient for RhoG. Mice transgenic for the OT-2 TCR specific for a peptide 323–339 of chicken ovalbumin presented by I-A[b] [18] and C57BL/6 bearing the pan-leukocyte marker allele CD45.1 were kindly provided by Dr. Carlos Ardavín (CNB, Madrid). B1-8[low] knock-in mice were kindly provided by Prof. Pavel Tolar (University College, London). These mice bear a pre-rearranged V region derived from the NP-binding antibody B1-8[44]. SWHEL mice are knock-in for a pre-rearranged V region and transgenic for the kappa light chain harboring specificity for hen-egg lysozyme (HEL)[45]. *Aicda*[−/−] mice[46] were used as negative controls for SHM assays and were kindly provided by Dr. Almudena R. Ramiro (CNIC, Madrid). All animals were backcrossed to the C57BL/6 background for at least 10 generations. For all in vivo experiments, age (6-10 weeks) and sex were matched. Animals from 6 to 16 weeks were used for in vitro experiments. Mice were maintained under SPF conditions in the animal facility of the Centro de Biología Molecular Severo Ochoa in accordance with applicable national and European guidelines. All animal procedures were approved by the ethical committee of the Centro de Biología Molecular Severo Ochoa.

**Cell purification**. Splenic B cells were negatively selected by staining with biotinylated anti-CD43 (clone S7) and anti-CD11b (clone M1/70) antibodies for 20 minutes on ice, washing and incubating the cells with streptavidin beads (Dynabeads M-280 Invitrogen), following the manufacturer's instructions. A Dynal Invitrogen Beads Separator was used to purify the cells. B1-8[hi] B cells were purified using biotinylated anti-CD43 (clone S7), anti-CD11b (clone M1/70) and anti-kappa (clone RMK-12) antibodies. OT-2 T cells from lymph nodes and spleen were purified using a mix of biotinylated antibodies: anti-B220 (clone RA3-6B2), anti-CD8 (clone 53-6.7), anti-NK1.1 (clone PK136), anti-CD11b, anti-GR1 (clone RB6-8C5), and anti-F4/80 (clone BM8). Splenic and lymph node B and T cells were maintained in RPMI 10% FBS supplemented with 2 mM L-glutamine, 100U/ml penicillin, 100 U/ml streptomycin, 20 μM β-mercaptoethanol, and 10 mM sodium pyruvate. Alternatively, plenic B cells and OT-2 T cells from lymph nodes were purified by negative selection using the CD43 (Ly-48) Microbeads (Miltenyi Biotec) and naïve CD4 + T cell Isolation Kits (Miltenyi Biotec), respectively. The cell suspension was then loaded onto a MACS column and a MACS Separator was used to purify the cells. All antibodies used are listed in Supplementary Table 1.

**Antigen-coated bead preparation**. To prepare beads with adsorbed antigen, a total of $130 \times 10^6$ carboxylated latex beads of 1 μm diameter (Polysciences) were incubated overnight with a concentration of 40 μg/ml of protein in 1 ml of PBS at 4 °C. When more than one protein was adsorbed, equimolar amounts were used. Beads were subsequently washed twice with PBS plus 1% BSA and 2 mM EDTA, and resuspended in RPMI medium. To prepare beads with covalently-bound antigen, the PolyLink Protein Coupling Kit (Polysciences) was used as indicated by the manufacturer. Ovalbumin (OVA) and hen-egg lysozyme (HEL) were purchased from SIGMA; NIP(15)-OVA and NP(25)-CGG from Biosearch Technology.

**Proliferation and stimulation assays**. The proliferation of OT-2 and B cells was assessed using CFSE, Cell Trace Violet (CTV), or Cell Trace Far Red (CTFR) labeling as specified by the manufacturer (Thermofisher). Proliferation index, defined as the total number of divisions divided by the number of cells that went into division, was obtained after analysis by flow cytometry (FACS Canto II) and FlowJo software. A total of $2 \times 10^5$ purified naïve B cells were co-cultured with purified OT-2 T cells at a 1:1 ratio together with antigen-coated beads or soluble antigen in a round-bottom 96-well plate. For the bead-bound stimulus, B cells were incubated with beads coated with NIP-OVA, NP-CGG, or HEL plus OVA at different bead:B cell ratios. For stimulation with soluble antigen, NIP-OVA or NP594 BSA were used at a concentration of 100 ng/ml. After different days of culture, cells were washed in PBS plus 1% BSA and stained for Tfh (CXCR5 dilution 1:100, PD1 dilution 1:200, Bcl-6 dilution 1:100), germinal center B cell (CD95 dilution 1:100, GL7 dilution 1:300, CD38 dilution 1:100, Bcl-6 dilution 1:100), plasma cells (CD138 dilution 1:100, Blimp-1 dilution 1:100), memory B cells (CD38 dilution 1:100, CD73 dilution 1:200, CD80 dilution 1:100, PDL2, dilution 1:100) and other (CD3, B220 dilution 1:100, IgM dilution 1:200, IgD dilution 1:200, IgG1 dilution 1:300, NP-PE dilution 1:100, Phalloidin) markers. Dead cells were excluded in all analyses by using either Ghost Dye 540 or Ghost Dye 780 (Tonbo). The intracellular stainings for Bcl-6 and Blimp1 were performed using the Foxp3/Transcription Factor Staining Buffer Set. Samples were analyzed by flow cytometry (FACS Canto II) and FlowJo software. Cultures were maintained in RPMI 10% FBS supplemented with 2 mM L-glutamine, 100 U/ml penicillin, 100 U/ml streptomycin, 50 μM β-mercaptoethanol, 10 mM sodium pyruvate, AANE, and 10 mM HEPES. After 4–5 days of culture, some cultures were supplemented with 1 ng/mL IL4 and 5 ng/mL IL21 (Peprotech). The culture supernatants were used to measure specific Igs by ELISA. For some experiments, B cells were also stimulated with 25 μg/ml lipopolysaccharide (LPS, Sigma-Aldrich) and 10 ng/ml IL4 (PreproTech) for 3 days.

**Measurement of antigen-specific antibodies**. To measure the release of NP-specific Ig in vitro, B cell: OT-2 T cell culture supernatants were incubated on NP(7)-BSA-coated or NP(41)-BSA-coated Costar p96 flat-bottom plates to measure the release of high- and low-affinity Igs, respectively. To quantify HEL-specific Ig, HEL was used for plate coating. The SBA Clonotyping System-HRP (Southern Biotech) was used to detect the presence of antigen-specific Ig isotypes. Absorbance at 415 nm was determined with an iMark Microplate Absorbance Reader (Bio-Rad). A mouse Immunoglobulin panel (SouthernBiotech) with defined concentrations of the different mouse Ig isotypes was used to quantify absolute concentrations of NP-specific antibodies. An independent standard curve was generated with serially diluted concentrations of each mouse antibody isotype assayed in goat anti-mouse Ig-coated wells. A 4-parameter-logistic curve generated with the absorbance data obtained was used for interpolations (GraphPad software).

**Confocal microscopy**. For antigen phagocytosis assays, purified B1-8[hi] B cells were incubated with NIP-OVA-coated fluorescent beads at a 1:3 B cell:bead ratio for 1 hour at 37 °C in RPMI plus 20 mM EDTA. Afterwards, cells were transferred to poly-L-lysine-treated coverslips and incubated with an anti-B220 and anti-Ovalbumin for 1 h at 0 °C. Subsequently, cells were fixed with 4% paraformaldehyde. For BCR-dependent signaling assays, starved B1-8[hi] cells were resuspended in RPMI + 0.2%BSA + 20 mM Hepes and incubated at 37 °C at different time points with 1 μm Y/G fluorescent beads coated with NIP-OVA at a 1:10 B cell:bead ratio. Cells were fixed in 4% paraformaldehyde at 0 °C for 20 minutes to stop stimulation, washed with Tris-buffered saline (TBS), and adhered to poly-L-lysine-treated coverslips. Extracellular staining for B220 was performed in TBS for 1 hour at 0 °C. After that, cells were stained for anti-phospho-Syk and antiphospho-Igα as suggested by the manufacturer (Cell Signaling). Coverslips were transferred to glass slides containing Mowiol. Confocal images were acquired with Zeiss LSM710 system. For analysis of T:B clusters, purified B1-8[hi] B cells were incubated with purified OT-2 T cells for 4 and 7 days in a six-well flat-bottom plate with 1 μm beads coated with NIP-OVA (1:3 B cell:beads ratio) or 100 ng/ml soluble NIP-OVA. Cells were fixed with 4% paraformaldehyde and transferred onto Poly-L-lysine-treated coverslips using a trimmed tip. Cells were stained for anti-CD4, -B220 and -GL7 in TNB buffer for 1 hour at RT. In order to monitor the degree of proliferation, cells were incubated with Cell Trace Marker or CFSE before set them in culture.

**BCR downmodulation and actin polymerization**. Purified B1-8[hi] B cells were incubated in a V-bottom 96-well plate with 1 μm NIP-OVA beads (1:3 cell:beads ratio) or 100 ng/ml soluble NIP-OVA. After spinning the cells, they were incubated at 37 °C for different time points. Subsequently, cells were washed in ice-cold PBS + 1%BSA and stained for anti-IgM and Phalloidin. Cells were analyzed by flow cytometry.

**NP saturation assay**. Purified B1-8[hi] B cells were placed in a V-bottom 96-well plate and incubated at 0 °C for 1 hour with different concentrations of 1 μm NIP-OVA beads or soluble NIP-OVA. Cells were washed and incubated with 2.5 μg/ml NP(36)-PE for 30 min at 0 C. After the last wash, cells were analyzed by flow cytometry.

**Microarray analysis of gene transcription**. For in vitro samples, purified B1-8[hi] B cells were set in culture with purified OT-2 T cells and stimulated for 3 days with 1 μm beads coated with NP-OVA (1:3 B cell:beads ratio) or 25 μg/mL LPS. For in vivo samples, mice were immunized i.p. with 200 μg NP-CGG complexed with alum for 7 days. Cells were recollected and spleens were harvested. Single-cell suspensions were stained for a live/death marker, anti-CD19, -GL7, and -CD95, as explained previously. GC B cells were sorted (CD19 + GL7 + CD95 + live/death−). Naïve follicular B cells were sorted based on the expression of CD19, CD23, and CD21. RNA from sorted cells was purified using an RNAeasy Plus Mini kit (QIAGEN). RNA integrity was assessed using an Agilent 2100 Bioanalyzer (Agilent). Labeling and hybridizations were performed according to protocols from Affymetrix. Briefly, 100 ng of total RNA were amplified and labeled using the WT Plus reagent kit (Affymetrix) and then hybridized to Mouse Gene 2.0 ST Array (Affymetrix). Washing and scanning were performed using an Affymetrix *GeneChip* System (GeneChip Hybridization Oven 645, GeneChip Fluidics Station 450, and GeneChip Scanner 7 G). Gene Set Enrichment Analyses were performed using GSEA v2.2.2.

**Immunoblot analysis of B-cell activation**. Purified B1-8[hi] B cells were resuspended in RPMI plus 20 mM Hepes and left in starving conditions for 1 hour. Cells were stimulated at different time points with NIP-OVA bound beads (ratio 3:1 beads/B cell) or 100 ng/mL soluble NIP-OVA. After stimulation, cells were lysed in Brij96 lysis buffer containing protease and phosphatase inhibitors (1% Brij96, 140 mM NaCl, 10 mM Tris–HCl [pH 7.8], 10 mM iodoacetamide, 1 mM PMSF, 1 μg/ml leupeptin, 1 μg/ml aprotinin, 1 mM sodium orthovanadate, 20 mM sodium fluoride and 5 mM of $MgCl_2$). Immunoblotting was performed as described previously[47].

**Somatic hypermutation**. Sanger sequencing: B1-8$^{hi}$ naive B cells and stimulated B cells after 7 days of culture with OT-2 CD4 + T cells plus NIP-OVA beads or soluble NIP-OVA were purified by cell sorting and their genomic DNA was extracted using QIAamp DNA kit (QIAGEN). B1-8V$_H$ genes were amplified by PCR with the Expand High Fidelity (Roche) and the primers forward 5′-CCATGGGATGGAGCTGTATCATCC-3′ and reverse 5′-GAGGAGACTGTGA-GAGTGGTGCC-3 as described previously[19]. PCR products were subcloned into PCR2.1 vector (Invitrogen) and individual clones were selected for sequencing by classical Sanger methods. Sequence analysis was performed using SeqMan software (DNAStar). PCR-sequencing (NGS): B1-8$^{lo}$ naive B cells and stimulated B cells after 5 days of culture with OT-2 CD4 + T cells plus NIP-OVA beads or soluble NIP-OVA were purified by cell sorting and their genomic DNA was extracted. B cells from B1-8$^{hi}$ mice and Aicda$^{-/-}$ mice stimulated for 3 days with LPS plus IL4 were used as positive and negative technical controls, respectively. The IgH Sμ region was PCR amplified using the following primers: forward 5′- AATGGA-TACCTCAGTGGTTTTTAATGGTGGGTTTA-3′ and reverse 5′-GCGGCCCGGCTCATTCCAGTTCATTACAG-3′ using Q5 High-Fidelity DNA Polymerase (New England Biolabs) for 27 cycles (95 °C for 30″, 60 °C for 30″, 72 °C for 60″). The B1-8V$_H$ region was amplified using the same protocol with the primers described above. PCR products were purified and fragmented using a sonicator (Covaris), and libraries were prepared according to the manufacturer's instructions (NEB Next Ultra II DNA Library Prep; New England Biolabs). Sequencing was performed paired-end (PE 2×250) in a MiSeq-v2 500 platform (Illumina). Analysis was performed by the Genomics and NGS Core Facility at the Centro de Biología Molecular Severo Ochoa (CBMSO, CSIC-UAM). High-quality reads (Phred Quality Score >20) were aligned using BWA aligner and were processed with SAMtools and custom scripting. Mutation count was done at C/G pairs within WRC̱Y/RG̱YW (W = A/T; R = C/G; Y = C/T) AID mutational hotspots.

**Adoptive transfer and immunizations**. In vitro-generated germinal center B cells or naive B cells ($10 \times 10^6$ cells per mouse) were injected intravenously into sublethally irradiated (6 Gy) 8-week-old C57BL/6 J mice. Thirty days after inoculation, mice were immunized intraperitoneally with 200 μg of NIP-OVA complexed with 100 μl of alum diluted 1:1 in PBS. After 7 days, spleens were collected and homogenized for analysis by flow cytometry.

**Statistics and reproducibility**. Statistical parameters, including the exact n value, and the mean ± S.D. or S.E.M., are described in the Figures and Figure legends. Parametric Student's $T$ tests and Two-Way ANOVA tests, were used to assess the significance of the mean differences as indicated. Experiments were repeated independently as described in the figure legends. The number of mice used for comparison was calculated from the preliminary experiments with the aim of generating significant data to give an alpha = 0.05 and a standard deviation of ~0.3 when a two-sided $T$ test was used. The different deviation of the control and test groups suggested the use of different numbers of each animal for the definitive experiments. Western blotting was performed at least twice with similar results and one representative image is shown in the figures. All the data were analyzed using the GraphPad Prism 7 software.

**Reporting summary**. Further information on research design is available in the Nature Portfolio Reporting Summary linked to this article.

## Data availability

All data is available as supplementary material and data files. The source data for all Figures is present in Supplementary Data 2. DNA expression data are present in Supplementary data 1. All other data are available from the corresponding author on reasonable request.

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

## Acknowledgements

We thank Nuria Martínez-Martín and Facundo Batista for critical reading of the manuscript. We are also indebted to Cristina Prieto, Valentina Blanco, and Tania Gómez for their expert technical assistance and Belen de Andrés and Maria Angeles Muñoz-Alcalá for helpful comments and advice on methodology. This work was supported by grants SAF2016-76394-R and PID2019-104935RB-I00 from the 'Comision Inter-ministerial de Ciencia y Tecnología' and by the European Research Council ERC 2013-Advanced Grant 334763 "NOVARIPP", (and from the Fundación Ramón Areces (to the CBMSO). M.T. is funded by the Biotechnology and Biological Sciences Research Council and Wellcome Foundation.

## Author contributions

Conceptualization, A.M.-R. P.D., and B.A.; methodology, A.M.-R., P.D., R.T., S.B., E.R.-B., and P.M.; software, D.A.; formal analysis, B.A.; Investigation, A.M.-R., R.T., S.B., P.M., C.L.O., and E.R.-B.; writing-original draft, B.A.; writing-review & editing, A.M.-R., P.D., R.T., C.L.O., and M.T.; funding acquisition, B.A., and M.T.; resources, B.A. and M.T.; supervision, B.A.

## Competing interests

The authors declare no competing interests.
