## [Peer Review File · Communications Biology]

Reviewers' comments:

Reviewer #1 (Remarks to the Author):

Martinez-Riano et al. describe a new in vitro culture set-up that can be used to study the interaction of antigen-specific B cells and T cells mimicking certain processes within the germinal center. The authors use this set-up to demonstrate that antigen-coated beads induce more robust B cell responses than soluble antigen and that bystander B cells that are not antigen-specific do not receive efficient help from T cells. I believe this experimental set-up to be very useful to the community of researchers interested in B and T cell biology since it allows to test different hypothesis concerning antigen-presentation, uptake, B and T cell communication.

My only major complain is that in the last part of the manuscript bystander cells are treated with soluble rather than bead bound antigen. In the first part of the paper the authors have shown that soluble antigen does not activate the cells in a similar fashion as bead bound antigen. Hence in the last part of the study bystander cells might not be activated sufficiently because they are unable to utilize T cell help but simply because the mode of activation coming from the antigen itself is rather weak. Here experiments should be performed where the bystander cells receive bead bound antigen comprised of a non-ova peptide.

Minor issues:

- Page 2.: Typing mistakes: "successful vaccines rely on activating a functional humoral with the generation..." The word "immune response" seems to be missing?
- Page 5, line 122: "...control soluble instead of bead-bound" The word "form" seems to be missing?
- Page 8, line 223: Typing mistake: "The the decline..."
- Page 415: I would not call CD73 a memory B cell marker. While it is present on a subset of memory B cells, CD73 can also be found on germinal center B cells in general (See Conter et al. 2014)
- In the material and methods section some important information seems to be missing - for example how the immunomagnetic depletion was done- what beads were used. Also, the clone and company name for antibodies seem to be missing. The authors don't seem to have filled out the reporting summary.
- In the acknowledgements C.L.O seems to be only listed for reviewing and editing the manuscript, editing manuscripts usually does not warrant authorship on a scientific paper
- References throughout the manuscript are numbered, but listed alphabetically in the reference list making it impossible to find the referenced paper
- In figure 1C an example staining should be shown, ideally with unstimulated T cells for comparison
- Fig.1E: It would be helpful if the number of NP labelling for high and low affinity antibodies was listed in the figure legend.
- Fig.1F: Are these ungated cells? Or cells gated on a B cell marker? It would be more helpful to show the percentage of IgG1+ cells in the population of B cells rather than all cells
- Figure 2 D and E: The authors should state in the figure legend how often the experiments were performed. If these experiments were performed only two times as stated in the material and methods section, the authors should repeat the experiments and either summarize them in a graph or include the repeats in the supplementary results. Judging by Akt phosphorylation in D and E, there seems to be great variability from experiment to experiment, hence careful repetition is necessary in my opinion.
- In figure 3, the number of independent repeats of the experiments should be stated in the figure legend.
- Page 27, line 865: Typing mistake?
- The authors should mention somewhere in the manuscript that not all B cells from SWHEL mice actually bind HEL. This information might be important for the reader in order to understand the strength of the observed B cell activation.

Reviewer #2 (Remarks to the Author):

a) Summary of the key results

The manuscript by Martínez-Riaño et al. describes a novel in vitro culture system to study germinal center (GCs) reactions. Based on their findings in 2018 (DOI 10.15252/embr.201846016) that follicular B cells can phagocytose large antigen-coated particles, they now demonstrate that BCR-mediated phagocytosis of bead-bound antigen by antigen-specific B cells together with cognate T cells can be used in vitro to induce robustly proliferating B cells that undergo isotype switching, produce high-affinity antibodies, and acquire a cell surface phenotype that resembles features of GC B cells. Acquisition of antigen through a BCR-dependent phagocytic process was linked to stronger and more sustained BCR signaling as compared to soluble antigen stimulation that resulted in GC B cells undergoing affinity maturation and somatic hypermutation regardless of the original affinity of the BCR. Although the manuscript describes a novel in vitro GC system utilizing the NP-OVA beads with only two cell types involved, several aspects need to be reconsidered.

b) Suggested improvements:

Major points:

1. Do the in vitro generated GC B cells also function in vivo? In this regard, the authors should discuss potential advantages over previously published systems such as Nojima cultures that did not require T cells but accessory cells providing survival/stimulatory signals (<https://pubmed.ncbi.nlm.nih.gov/21897376/>). Nevertheless, they were functional in vivo. A potential in vivo experiment could be for example in which GC B cell-deficient mice (e.g. AID^{-/-}) are immunized with NP-Ova and then the in vitro GC B cells are transferred and tested.
2. The input cells for the in vitro GC system were ovalbumin (OVA)-specific OT-II TCR-transgenic CD4⁺ T cells, of which around 50-70% differentiated into CXCR5⁺PD-1⁺ T cells after 4 days of co-culture together with NP-specific B1-8 high BCR-transgenic B cells (Fig 1c or Fig 8e). Nevertheless, no primary flow cytometry data is shown. Generation of mouse Tfh cells in vitro is a controversial topic (Crotty, Immunity 2019), thus sample plots should be shown here as well. As it is unclear if B cells (that express high levels of CXCR5) and dead cells (unspecific staining) were excluded from the analysis, the authors should also provide the gating strategy of follicular T helper cells.
3. In line with the point above, it is unclear if dead cells were excluded from any downstream analyses of the B cells (apart from the micro array experiment, where it is specifically stated). Dead cells are known to have greater autofluorescence and increased non-specific antibody binding leading to false positive results and misinterpretation of data. Was a viability dye included in the staining protocol to distinguish live from dead cells in the in vitro experiments?
4. The dead cell exclusion is also critical for the T-B conjugate experiments presented in Fig 8 and suppl. Fig S4. T-B cell conjugate detection is not trivial and controversial in the field. It would be particularly important to exclude dead cells in these experiments. As another control: What happens if EDTA is present in the flow buffer, as most commonly done for flow cytometry experiments? Would the conjugates disappear?
5. This manuscript mainly focuses on the B cells of the co-culture system. As already alluded to above, a better characterization of the input OT-II CD4⁺ T cells would provide important insights into the role of these cells in the in vitro GC system. Are they really Tfh cells, i.e. do they express Bcl6 protein (similar to the GC B cells shown in Fig 4c)? Are B cells excluded from all analyses of Tfh cells?

Minor Points:

6. The redundant analysis of GC formation of low-affinity B1-8low B cells stimulated with bead-bound antigen and cognate T cells (Fig 5) can be moved to the supplement.
7. When depicting sample data, it is better to show plots that represent the overall findings rather than the outliers (e.g. Fig. 3a)
8. There are a few minor mistakes in the manuscript text. For example: Line 27: Successful vaccines

rely on activating a functional humoral "immune response" with the generation of [...]. Line 29: A major hurdle "to understand" the mechanisms of B cell:T cell cooperation [...]. Line 47: B cells have to recognize cognate antigen via their B cell antigen receptor (BCR), "internalize and present it" [...]. Line 306: (Fig. 6c).

9. "FACS" is a trademark owned by BD. Unless it is referring to a machine built by BD, the term should be avoided and replaced with "flow cytometry".

Reviewer #3 (Remarks to the Author):

In the current study, Alarcon et al. showed that stimulation of B cells by antigen-coated microbeads led to the formation of GC-like clusters in a co-culture system with cognate CD4 T cells. These beads were phagocytosed by B cells, and induced prolonged B cell activation compared to soluble antigen. The authors further showed that such phagocytic antigen delivery encouraged class-switching and somatic hypermutation in B cells. It is in general an interesting finding and could potentially help recapitulate GC response in vitro. However, several clarifications need to be done.

Major concerns:

1. The first result section is misleading. The authors claim that the in vitro system they set up generates "antigen-specific class switched antibodies of high affinity", hinting on possible affinity maturation. However, such culture was started with high affinity B1-8hi cells, and there is a lack of evidence for the evolution of antibody affinity. It is worth noting that by doing ELISA with NP7- and NP41-BSA, the authors are indeed measuring high affinity and total anti-NP antibodies, rather than high and low affinity antibodies as mentioned in the text.
2. Related to the first point, although the authors showed that their in vitro system supported CSR and SHM, they failed to test clonal competition and affinity maturation, two important hallmarks of GC reaction. What happens if B1-8hi and B1-8lo cells were put into the same culture? Were affinity-enhancing mutations selected?
3. It is mentioned in the discussion section that RhoG does not lead to blockade of B cell phagocytosis in vitro, and therefore it is confusing to me why the authors used RhoG deficient cells to probe the role of phagocytosis in the enhancement of BCR signaling.
4. In the bystander experiment described in Figure 9, the authors concluded that bystander cells form premature memory B cells. However, to reach such conclusion, the authors need to enumerate all divided cells in the culture at different time points, to rule out that the increase percentage of CD38+ cells was not due to the death of divided CD38- cells. From the FACS profile shown in figure 9b, it looks like there was a loss of CTRlo cells at later time points of the culture.

Minor points:

1. The authors need to show FACS profiles for Figure 1c, 8f, 8g.
2. The blots in Figure 2e seem to be over-developed.
3. Figure 2f is confusing—the zoomed area should be boxed.
4. The images shown for the "soluble NIP-OVA" group in Figure 3a is not at all representative, judged by the statistical summary shown on the right.
5. It is not clear what the blue portion in the pie chart indicates in Figure 6b.

Point-by-point response to Reviewers' comments:

Reviewer #1

We thank the Reviewer for the positive view on the relevance of our work.

-Here experiments should be performed where the bystander cells receive bead bound antigen comprised of a non-ova peptide.

The point of the Reviewer is well taken and we agree that this is a control experiment we needed to undertake. We have now carried out control experiments to determine if bystander B cells do undergo class-switching and produce mature immunoglobulins if a bead-bound rather than soluble antigen is provided. The results are in supplemental Figure S5a and S5b. Our interpretation of the data is that the different response of cognate and bystander B cells is not due in this case to the format of the antigen but to the fact that bystander B cells cannot present OVA antigen to OT2 T cells.

Minor issues:

- Page 2.: Typing mistakes: “successful vaccines rely on activating a functional humoral with the generation...” The word “immune response” seems to be missing?

Yes, it was a mistake. It is now corrected and shown in blue type.

- Page 5, line 122: “...control soluble instead of bead-bound” The word “form” seems to be missing?

Yes, it is now corrected and shown in blue type.

- Page 8, line 223: Typing mistake: “The the decline...”

This has been corrected

- Page 415: I would not call CD73 a memory B cell marker. While it is present on a subset of memory B cells, CD73 can also be found on germinal center B cells in general (See Conter et al. 2014)

We agree with the Reviewer, but it is also a memory marker useful to define memory subsets. Combined with high CD38 expression and with the fact that we measure it in divided cells, we think we can be quite sure it is helping to define memory B cells.

- In the material and methods section some important information seems to be missing - for example how the immunomagnetic depletion was done- what beads were used. Also, the clone and company name for antibodies seem to be missing. The authors don't seem to have filled out the reporting summary.

We have now included in Methods a description on how negative selection for B cells and T cells was carried out. We now also provide a list of reagents in the reporting summary.

- In the acknowledgements C.L.O seems to be only listed for reviewing and editing the manuscript, editing manuscripts usually does not warrant authorship on a scientific paper

C.L.O. also contributed, although to smaller extend, to the generation of data. This is now included in the acknowledgements section.

- References throughout the manuscript are numbered, but listed alphabetically in the reference list making it impossible to find the referenced paper

We have now corrected this mistake

- In figure 1C an example staining should be shown, ideally with unstimulated T cells for comparison

We now show an example of staining in Fig. 1c. In fact, we show a difference in PD1 and CXCR5 expression by divided and non-divided cells (the latter as a control).

- Fig.1E: It would be helpful if the number of NP labelling for high and low affinity antibodies was listed in the figure legend.

We now indicate the degree of substitution of BSA by NP molecules in the figure legend.

- Fig.1F: Are these ungated cells? Or cells gated on a B cell marker? It would be more helpful to show the percentage of IgG1+ cells in the population of B cells rather than all cells

We apologize for the omission, the analysis was on gated B220+ B cells. This is now mentioned in the legend to the figure.

- Figure 2 D and E: The authors should state in the figure legend how often the experiments were performed. If these experiments were performed only two times as stated in the material and methods section, the authors should repeat the experiments and either summarize them in a graph or include the repeats in the supplementary results. Judging by Akt phosphorylation in D and E, there seems to be great variability from experiment to experiment, hence careful repetition is necessary in my opinion.

The western blot experiments were carried out in triplicate in parallel. Only one blot per experiment is shown but the quantification of the band intensities is now presented as mean \pm s.e.m. in the graphs plots in the Figure panels.

- In figure 3, the number of independent repeats of the experiments should be stated in the figure legend.

Cluster formation has been witnessed dozens of times, basically in every experiment of GC formation. We now indicate the number of times in which cluster sizes were measured.

- Page 27, line 865: Typing mistake?

It was a mistake. Thank you very much!

- The authors should mention somewhere in the manuscript that not all B cells from SWHEL mice actually bind HEL. This information might be important for the reader in order to understand the strength of the observed B cell activation.

We thank the Reviewer for that piece of advice. We now show in Fig. S5a that 35% of the SWHEL naïve B cells actually bind HEL.

Reviewer #2

We thank the Reviewer for the kind evaluation and critical contribution to our work.

Major points:

1. Do the in vitro generated GC B cells also function in vivo? In this regard, the authors should discuss potential advantages over previously published systems such as Nojima cultures that did not require T cells but accessory cells providing survival/stimulatory signals

(<https://pubmed.ncbi.nlm.nih.gov/21897376/>). Nevertheless, they were functional in vivo. A potential in vivo experiment could be for example in which GC B cell-deficient mice (e.g. AID^{-/-}) are immunized with NP-Ova and then the in vitro GC B cells are transferred and tested.

We have now carried out an experiment similar to the one suggested by the Reviewer in order to demonstrate that the in vitro generated GC B cells are functional in vivo. However, we have preferred not to use immunodeficient mice such as AID^{-/-} as recipients but rather use fully immune competent mice bearing a different CD45 allele to distinguish endogenous from transferred cells. We show in Supplemental Fig. S4 that transferred day 5 B cells generated in vitro do respond to stimulation with antigen in vivo and they generate more class-switched antigen-specific B cells than the corresponding control cells consisting of transferred naïve B cells.

2. The input cells for the in vitro GC system were ovalbumin (OVA)-specific OT-II TCR-transgenic CD4⁺ T cells, of which around 50-70% differentiated into CXCR5⁺PD-1⁺ T cells after 4 days of co-culture together with NP-specific B1-8 high BCR-transgenic B cells (Fig 1c or Fig 8e). Nevertheless, no primary flow cytometry data is shown. Generation of mouse Tfh cells in vitro is a controversial topic (Crotty, Immunity 2019), thus sample plots should be shown here as well. As it is unclear if B cells (that express high levels of CXCR5) and dead cells (unspecific staining) were excluded from the analysis, the authors should also provide the gating strategy of follicular T helper cells.

We always include a dead/live cell marker, generally Ghost Dye, to eliminate dead cells from all our analyses. An example of how Tfh cells are gated according to the expression of CXCR5 and PD1 is now shown in Fig. 1c. In addition, we now show in Fig. 7f that the gated Tfh cells, according to CXCR5 and PD1, are also Bcl6 positive, thus indicating that they are truly Tfh.

3. In line with the point above, it is unclear if dead cells were excluded from any downstream analyses of the B cells (apart from the micro array experiment, where it is specifically stated). Dead cells are known to have greater autofluorescence and increased non-specific antibody binding leading to false positive results and misinterpretation of data. Was a viability dye included in the staining protocol to distinguish live from dead cells in the in vitro experiments?

Yes, we always included a dead/live cell marker in order to exclude dead cells. This is now specified in the Methods section.

4. The dead cell exclusion is also critical for the T-B conjugate experiments presented in Fig 8 and suppl. Fig S4. T-B cell conjugate detection is not trivial and controversial in the field. It would be particularly important to exclude dead cells in these experiments. As another control: What happens if EDTA is present in the flow buffer, as most commonly done for flow cytometry experiments? Would the conjugates disappear?

Dead cells were excluded as indicated above and the flow buffer consisted of PBS with no bivalent cations and with 1 mM EDTA. So, the conjugates detected in the cytometer are quite resistant to disruption.

5. This manuscript mainly focuses on the B cells of the co-culture system. As already alluded to above, a better characterization of the input OT-II CD4⁺ T cells would provide important insights into the role of these cells in the in vitro GC system. Are they really Tfh cells, i.e. do they express Bcl6 protein (similar to the GC B cells shown in Fig 4c)? Are B cells excluded from all analyses of Tfh

cells?

Staining with Bcl6 is now shown in Fig. 7f. The CXCR5+PD1+ CD4+ T cells are clearly positive for Bcl6, so we are quite confident in that they are truly Tfh. B cells are always excluded using CD19 as a B cell marker.

Minor Points:

6. The redundant analysis of GC formation of low-affinity B1-8low B cells stimulated with bead-bound antigen and cognate T cells (Fig 5) can be moved to the supplement.

We have now moved the analysis of GC formation by B1-8low B cells to the supplemental material (Fig. S2) as suggested by the Reviewer. This was a good idea because reduces the number of main figures

7. When depicting sample data, it is better to show plots that represent the overall findings rather than the outliers (e.g. Fig. 3a)

We agree with the Reviewer. We now show in Figure 3a, micrographs of clusters that represent the average of their size. We have replaced the example for soluble NIP-OVA stimulation.

8. There are a few minor mistakes in the manuscript text. For example: Line 27: Successful vaccines rely on activating a functional humoral "immune response" with the generation of [...]. Line 29: A major hurdle "to understand" the mechanisms of B cell:T cell cooperation [...]. Line 47: B cells have to recognize cognate antigen via their B cell antigen receptor (BCR), "internalize and present it" [...]. Line 306: (Fig. 6c).

We thank the Reviewer for the careful revision of the manuscript. We have now done all corrections and shown them in blue type

9. "FACS" is a trademark owned by BD. Unless it is referring to a machine built by BD, the term should be avoided and replaced with "flow cytometry".

Agree. We have now replaced the term

Reviewer #3

We thank the Reviewer for the positive opinion about our paper. We agree that this will become a method used by many others to study germinal center formation mechanisms.

Major concerns:

1. The first result section is misleading. The authors claim that the in vitro system they set up generates “antigen-specific class switched antibodies of high affinity”, hinting on possible affinity maturation. However, such culture was started with high affinity B1-8hi cells, and there is a lack of evidence for the evolvement of antibody affinity. It is worth noting that by doing ELISA with NP7- and NP41-BSA, the authors are indeed measuring high affinity and total anti-NP antibodies, rather than high and low affinity antibodies as mentioned in the text.

We have corrected “low affinity” by “Total” Igs in Fig. 1e. B1-8hi B cells do express a “high” affinity BCR but that does not mean that the affinity cannot be even higher. We show that B1-8hi cells undergo hypermutation and that there is a selection of sequences with amino acid replacements that affect the three CDRs of the heavy chain or their vicinity (Fig. S3). In addition, we show now evidence of affinity maturation as explained in response to the second major concern.

2. Related to the first point, although the authors showed that their in vitro system supported CSR and SHM, they failed to test clonal competition and affinity maturation, two important hallmarks of GC reaction. What happens if B1-8hi and B1-8lo cells were put into the same culture? Were affinity-enhancing mutations selected?

This is a good point that was already partly answered in the original version of the manuscript and that now we hope to have improved. We already showed that an estimate of the affinity of the secreted anti-NP antibodies shows that class-switched ones (IgG1 and IgG2a) have more affinity than IgM. This already indicates affinity maturation. Now, we have repeated experiments for the analysis of how B1-8low B cells increase their affinity (bind better NIP at low concentrations) as they experience class switching (IgD- vs IgD+) and also in comparison to naïve cells (Fig. 5e). We also show that as B1-8low B cells divide in the in vitro system, their capacity to bind low or very low concentrations of NIP antigen increases (Fig. 5f). This again suggests that there is a process of affinity maturation that begins with cell division.

3. It is mentioned in the discussion section that RhoG does not lead to blockade of B cell phagocytosis in vitro, and therefore it is confusing to me why the authors used RhoG deficient cells to probe the role of phagocytosis in the enhancement of BCR signaling.

We used RhoG-deficient mice for two reasons: we have already demonstrated that they impair antigen-triggered B cells in vivo (Martinez-Riano, EMBO Rep 2018; Ref. 8), and because we did not find a better and cleaner candidate gene.

4. In the bystander experiment described in Figure 9, the authors concluded that bystander cells form premature memory B cells. However, to reach such conclusion, the authors need to enumerate all divided cells in the culture at different time points, to rule out that the increase percentage of CD38+ cells was not due to the death of divided CD38- cells. From the FACS profile shown in figure 9b, it looks like there was a loss of CTRlo cells at later time points of the culture. We cannot completely exclude the possibility posed by the Reviewer that bystander CD8low cells are dying after day 3 (Fig. 8c and Fig. S7b). However, we can say that there is not a massive B cell death in the system and the total number of live cognate and bystander B cells remain comparable along the 7 days of culture (Fig. 8b). However, we think we can propose the early differentiation of bystander cells into memory B cells because after day 3, divided cells do express (regain) high levels of CD38 and because they also express the CD73 marker which together with high CD38 strongly suggest formation of a B cell memory subset (Fig. 8d).

Minor points:

1. The authors need to show FACS profiles for Figure 1c, 8f, 8g.

We have included FACS profile for Fig. 1c. Figure 8f is not longer included in the manuscript. Gating for cell conjugates of what is now Fig. 7g, is shown in Figure S6. Figures are quite packed already.

2. The blots in Figure 2e seem to be over-developed.

No, no they are not over-developed. We did not subtract all background and that is why it appears grey rather than white

3. Figure 2f is confusing—the zoomed area should be boxed.

This is now boxed as the Reviewer suggested

4. The images shown for the “soluble NIP-OVA” group in Figure 3a is not at all representative, judged by the statistical summary shown on the right.

We are now showing a more representative cluster of cells stimulated with soluble antigen.

5. It is not clear what the blue portion in the pie chart indicates in Figure 6b.

We now explained in the legend to what is now Fig. 5b: Blue segments in the pie charts are proportional to the number of sequences carrying mutations. The total number of independent sequences analyzed is indicated in the center of each chart. The calculated mutation frequency per base pair is indicated underneath.

REVIEWERS' COMMENTS:

Reviewer #1 (Remarks to the Author):

The authors have satisfactorily answered my questions and I think the additional experiment showing that antigen covered beads are not sufficient to enable bystander B cells to gather T cell help is very useful. Overall, I think the described system can be of great use to study germinal center processes in the future.

My only comment is that I remain of the opinion that I would not call B cells "memory B cells" solely based on their CD38 and CD73 expression. Both are expressed on other B cell subsets , sometimes together (CLL, multiple myeloma) and participate in the same adenosine generating pathway. Not much is known of this pathway in B cells and it cannot be excluded that these markers rise on exhausted cells, dying cells, dysfunctional cells, regulatory B cells or any other non memory B cell stage.

The experiments on their own are interesting, the discussion appears overreaching.

Reviewer #2 (Remarks to the Author):

The authors have addressed most of my concerns. There are still several grammatical errors in the text that should be fixed before publication.

Reviewer #3 (Remarks to the Author):

The authors have sufficiently addressed my concerns.